# Measuring extremes-driven direct biophysical impacts in agricultural drought damages

Mansi Nagpal[1], Jasmin Heilemann[1], Luis Samaniego[2,3], Bernd Klauer[1], Erik Gawel[1,4], and Christian Klassert[1]

[1] Department of Economics, Helmholtz Centre for Environmental Research–UFZ, 04318 Leipzig, Germany

[2] Department of Computational Hydrosystems (CHS), Helmholtz Center for Environmental Research – UFZ, 04318 Leipzig, Germany

[3] University of Potsdam, Institute of Environmental Science and Geography, Potsdam, Germany

[4] University of Leipzig, Faculty of Economics and Business Management, 04109 Leipzig, Germany

Correspondence to: Mansi Nagpal (mansi.nagpal@ufz.de)

**Abstract**

Assessing the economic implications of droughts has become increasingly important due to their substantial impacts on agriculture. Existing empirical analyses for drought damages are often conducted on a national scale without spatially distributed data, which might bias estimates. Furthermore, the cumulative effects of multiple weather extremes, such as heat or preceded frost co-occurring with drought, are often overlooked. Measuring the direct biophysical impacts of such extremes on agriculture is essential for more

precise risk assessment. This study presents a comprehensive economic impact assessment framework to measure the cumulative damages of droughts and other hydro-meteorological extremes on agriculture, focusing on eight major field crops in Germany. By utilizing a statistical yield model, we isolate the effects of multiple extremes on crop yields from other influencing factors (such as pests & diseases, farm management) and analyze their contribution to farm revenue losses during droughts at the district level from 2016-2022. Our findings indicate that the average annual direct biophysical damage caused by extremes under drought conditions

during this period amounts to € 781 million (sensitivity range: €766 million-€812 million) across Germany. The study also reveals that biophysical impacts of extremes alone account for 60% of reported revenue damages during widespread drought years. For maize, direct biophysical damage explains up to 97% (2018) of revenue losses. Additionally, comparison of national-level damage estimates using aggregated and spatially disaggregated data shows that the aggregated data matches overall results, but diverges for maize and wheat, highlighting the importance of spatially distributed damage assessment. In this paper, we provide detailed

estimates of extremes-driven direct biophysical damages at the district level, offering a high-resolution understanding of the spatial and temporal variability of these impacts. Assessing the extent of revenue losses resulting from these extremes alone can provide valuable insights for the development of effective drought mitigation programs and guide policy planning at local and national levels to enhance the resilience of the agricultural sector against future climate extremes. Future integration of routine drought damage estimation into operational monitoring and forecasting systems would enhance early warning capabilities, improve

economic preparedness against increasing weather extremes, and support more proactive adaptation strategies.

**Keywords:** Drought impacts, economic impacts, climate change adaptation, extreme events, Germany

## 1 Introduction

Recent decades have seen a significant change in global temperature and precipitation patterns (Daramola & Xu, 2022). As climate change progresses, extreme events such as droughts and heat waves are expected to increase (Samaniego et al., 2018). The impacts

of such hydro-meteorological extreme events on water resources and agriculture, which are strongly linked to global food security, are already being felt (Shukla et al., 2019). Quantifying the costs of these impacts and understanding their drivers is a prerequisite for assessing vulnerabilities and designing adaptation measures to increase the resilience of the agricultural sector (Rose, 2004).

A variety of factors including war (Appau et al., 2021), disease and pests (Savary et al., 2019), and extreme weather (Lesk et al., 2016) affect crop yields. Of these factors, climate variability has particularly pronounced impacts on yield variations. In major

agricultural production regions globally, over 60% of yield variability can be explained by climate variability (D. K. Ray et al., 2015). Drought, in particular, is one of the most severe climate-related hazards, significantly reducing crop yields and incurring high crop production losses. For instance, it is estimated that the average crop production impact of droughts (and heatwaves) has tripled from 1964 to 2015 across the European Union (Brás et al., 2021). Given the profound impact of droughts on agriculture, it is crucial to understand the economic consequences and the extent of damage caused by such extremes. However, the complexity

of drought occurrences—characterized by their slow development, spatial and temporal accumulation, and significant variability in severity and intensity—makes research on their economic impacts challenging (Eckhardt et al., 2019).

Droughts are periods of significantly reduced moisture levels in the Earth system (Wilhite & Glantz, 1985), leading to restrictions in water availability and causing detrimental impacts on various environmental systems and economic sectors. Generally, there are four types of droughts: meteorological droughts (precipitation deficit), agricultural droughts (soil moisture deficit), hydrological

droughts (abnormal streamflow, groundwater, reservoir, or lake deficits), and socioeconomic droughts (abnormal deficit due to imbalance between supply and demand) (Wilhite & Glantz, 1985).

The impacts of droughts extend to agriculture, livestock, forestry, energy, and industries, and even threaten human safety (de Brito et al., 2020). Due to its sensitivity to weather variability and soil moisture, the agricultural sector is often the first sector to be affected by drought  (Y. Ding et al., 2011; Wilhite, 2000). Agricultural droughts are soil moisture droughts that occur when crop

water requirements are not met during the growing season due to a reduced water supply in the soil, mainly caused by decreased precipitation or/and increased temperatures (X. Liu et al., 2016; Rakovec et al., 2022). This lack of moisture affects crop growth and yields, posing a significant threat to harvests. These impacts can lead to a substantial decline in crop revenues and/or an increase in production costs, ultimately reducing farm profits, affecting farmers' livelihoods and economic stability within the sector, and threatening food security (FAO, 2023; Ziolkowska, 2016).

The impact of drought on agricultural production is not solely determined by the severity of the drought itself, but also by exposure to different weather extremes throughout the growing season (Haqiqi et al., 2021; Peichl et al., 2018; Schmitt et al., 2022). For example, extreme heat during summer droughts can intensify damage to crops such as maize, further reducing yields (AghaKouchak et al., 2014). Similarly, winter crops like wheat can suffer significant losses from drought followed by periods of excessive rainfall, negatively affecting yields and harvest quality (J. Ding et al., 2018; Zampieri et al., 2017). Most research on

measuring the economic impacts of extreme events like droughts has been confined to assessing the impacts of specific weather extremes, despite growing evidence that such events are frequently driven by multiple interrelated climate drivers that can occur concurrently or successively within the same geographic area (AghaKouchak et al., 2014; Deng et al., 2024; Rakovec et al., 2022; Zscheischler et al., 2018, 2020). Failing to account for such concurrently or successively occurring extremes is likely to oversimplify the process leading to damages, underestimate the cumulative effects of weather extremes on crops, and may result

in an incomplete risk perception and inaccurate damage estimates (Meyer et al., 2013).

In this study, we address this bias by assessing the economic damage of drought in combination with concurrent or successive weather extremes in rainfed agriculture. The aim of this study is to measure the direct biophysical damage of extreme hydro-meteorological drivers during droughts (hereafter called *direct biophysically-induced damages*) and assess their contribution to farm revenue losses. These damages refer to the loss in revenue caused by the effects of extreme hydro-meteorological drivers on

crop yields, without accounting for other economic impacts, such as changes in costs. They include the effects of droughts themselves, as well as additional damage from concurrent or successive weather extremes that exacerbate drought-related effects in regions experiencing drought conditions. To isolate the biophysical impacts of these extremes on crop yields from other

influencing factors, we employ crop specific statistical yield models. By comparing the direct biophysically-induced damages estimated from these models with reported farm revenue losses, we can identify the relative contribution of these factors across different regions and crops, which can guide more targeted drought adaptation and enable better decision-making.

The empirical analysis of direct biophysically-induced damages during droughts is done at the district level for rainfed agriculture for eight major field crops in Germany from 2016-2022. These estimates are derived from the methodology used to measure the damages of the 2018 and 2019 droughts in Germany (Trenczek et al., 2022). We have enhanced this methodology for our current assessment.

Additionally, we demonstrate the utility of high-resolution damage assessment by comparing damages at the national level derived using both national-level and regional-level data. Existing research on measuring the damages of droughts on agriculture often focuses on national-level damage assessments without considering spatially distributed data and typically examines specific drought events (COPA-COGECA, 2003; Trenczek et al., 2022). Alternatively, there are several empirical studies analysing drought damages at the farm level that often incorporate adaptation strategies (van Duinen et al., 2015; Wens et al., 2021), input changes (Prasanna, 2018) and factors affecting localized responses to droughts (Ahmad et al., 2022; Garbero & Muttarak, 2013; Gray et al., 2009). Their findings are tailored to specific context and may not be readily scalable to broader regions. Conversely, national-level assessments, though comprehensive, fail to capture the spatial variability of drought impacts. As droughts can vary greatly across different locations and times (Jaeger et al., 2013; Samaniego et al., 2013), there is a need for consistent, spatially-explicit damage assessments (Meyer et al., 2013) bridging the gap between farm-level-detail and national-level scope. Our analysis reveals that high-resolution damage assessment using regional-level data provide a more accurate quantification of crop-specific damages, which might not be captured by assessments using national-level data.

This study offers detailed, high-resolution estimates of extremes-driven direct biophysically-induced damages at the district level, offering insights into the spatial and temporal variability of these impacts. By accounting for concurrent or successive weather extremes alongside droughts, our research provides a more accurate assessment of revenue losses during droughts. These findings can inform the development of effective drought mitigation programs and guide policy planning at local and national levels to enhance the resilience of the agricultural sector against future climate extremes.

## 2 Methodology

### 2.1 Overview of Analytical Approach

This study focuses on isolating the direct biophysically-induced damages of weather extremes during droughts on agriculture from other influencing factors and assessing their contribution to farm revenue losses. **Figure 1** outlines our approach to quantifying these damages in rainfed agriculture by illustrating both the causal pathways by which droughts and related extreme events lead to revenue losses during the year the drought occurs and the empirical methods and data used to measure them. In our analysis, we quantify the direct biophysical impacts of concurrently or successively occurring weather-extremes (rather than changes in mean temperature, precipitation, etc. as done in context of climate change) on crop yields and the resulting damages.

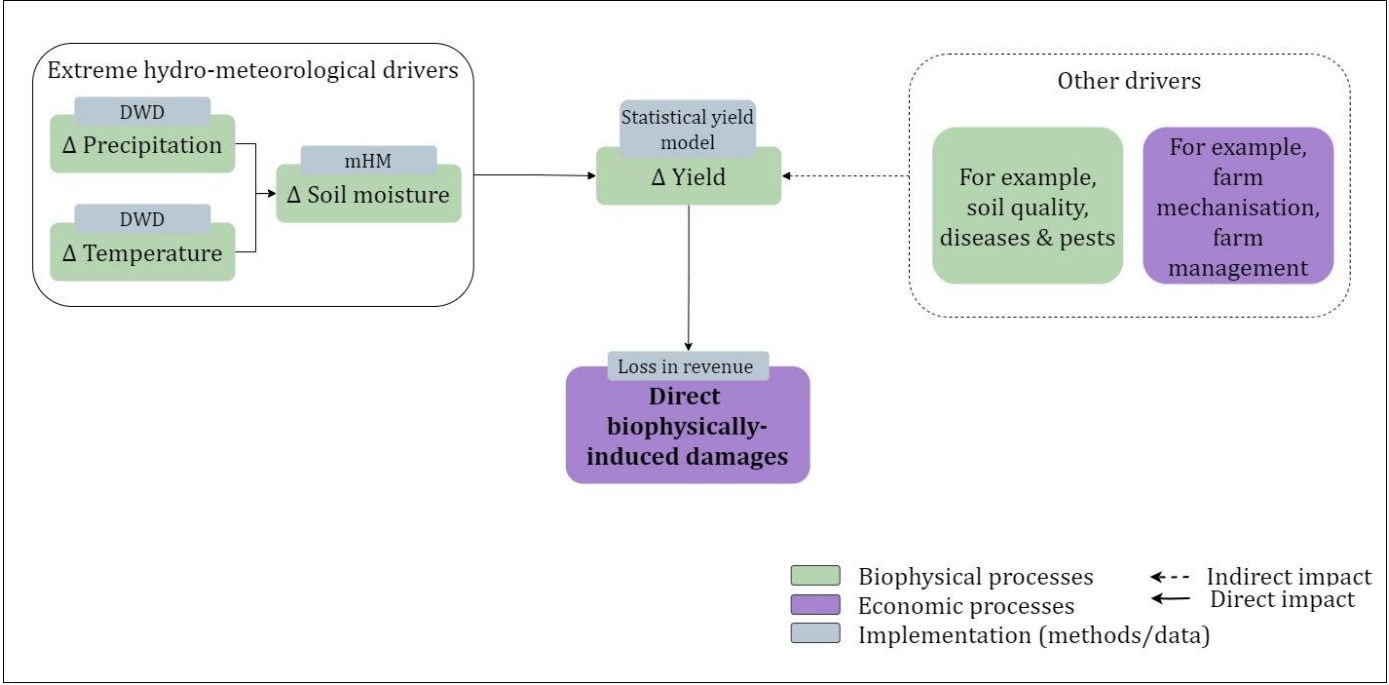

**Figure 1 Schematic illustration of the approach for quantifying the direct biophysically-induced damages, measured as the farmers' revenue losses due to hydro-meteorological extremes during droughts. Temperature and precipitation data from the German Weather Service (DWD) and soil moisture estimates from the mesoscale Hydrological Model (mHM) serve as key hydro-meteorological inputs in**

**a statistical yield model to estimate the direct impact of drought and related extremes on crop yields. These effects are isolated from other influencing factors, such as soil quality and farm management, to focus on the direct biophysical drivers of yield losses.**

Agricultural droughts occur when soil moisture levels are insufficient to meet crop water requirements during the growing season, making soil moisture (anomalies) a more accurate predictor of biophysical impacts than precipitation or temperature alone (Bachmair et al., 2016). The importance of soil moisture in informing agricultural damage assessment is increasingly recognized

(Haqiqi et al., 2021; Peichl et al., 2018). Declining soil moisture due to drought directly impedes crop growth and reduces crop yields, which are referred to as the direct impacts of droughts on agriculture (Meyer et al., 2013). These biophysical impacts can be exacerbated by the occurrence of other weather extremes. For example, heat, can exacerbate damage to summer-grown crops like maize during droughts (AghaKouchak et al., 2014; Haqiqi et al., 2021),while  excessive wet conditions during the growing season in addition to drought can lead to substantial damage to winter-grown crops like wheat (Ben-Ari et al., 2018; Zampieri et

al., 2017). Thus, under evolving climate conditions, it is crucial to assess the direct biophysical impacts of droughts in conjunction with various hydro-meteorological extremes, as these factors collectively have been shown to explain a significant proportion of crop yield variability (Schmitt et al., 2022; E. Vogel et al., 2019; Webber et al., 2020). In our analysis, the direct biophysical effect of extreme hydro-meteorological drivers on crop yield is estimated using crop-specific statistical yield models using temperature, precipitation and soil moisture as key input variables (detailed in Sect. 2.3). As shown in **Figure 1**, the soil moisture data, which

are central for defining drought conditions, is derived using the mesoscale Hydrological Model (mHM) (Samaniego et al., 2010). Additionally, the sources of temperature and precipitation data (German Weather Service, DWD) are also indicated for consistency. The impact of declining soil moisture because of drought is more pronounced in rainfed agriculture, where crop yields can be significantly affected in the short term (Kurukulasuriya et al., 2006). Conversely, irrigation helps buffer the impact of low soil moisture on crop yields. However, if the drought persists and leads to acute water shortage and competition for water use by other

users, it can still cause considerable damage to irrigated agriculture during droughts (Smith & Edwards, 2021).

While the analysis presented in this paper focuses on measuring direct biophysically-induced damages in rainfed agriculture, it is important to note that the impacts of droughts extend beyond these direct effects. For completeness, **Supplementary Figure S1** presents the broader economic impacts of drought and related weather extremes within a drought year, incorporating both rainfed and irrigated agriculture. These economic impacts arise because the direct damage to crop yields by drought and other weather extremes sets in motion a series of economic processes (Diaz & Moore, 2017). The biophysical impact on crop yields results in a decrease in harvest that leads to negative supply shocks which can raise the prices of agricultural products. These price increases are known as indirect impacts of droughts and must be considered in economic impact assessments (Y. Ding et al., 2011; Rose, 2004). Beyond these indirect impacts, farmers may implement various short-term risk mitigation strategies, such as adjusting their inputs or employing supplemental irrigation, to lessen the impact of the drought. These strategies, however, come with associated costs that need to be considered when estimating drought damages and are referred to as adaptation costs. The total economic impact of droughts and related extreme events (referred to as *extremes-driven impacts*) thus encompass indirect impacts on prices, changes in inputs and costs, in addition to the direct damages. In some cases, farmers may benefit from higher prices if the percentage increase in price exceeds the decrease in supply. This is particularly profitable for farmers operating outside the drought-affected area or farmers using irrigation. However, such impacts are difficult to measure using only national data and may require more detailed spatial assessments at the regional level. Moreover, given that droughts are unevenly distributed over regions, it is important to incorporate sufficiently detailed spatial disaggregation to assess the economic impacts on a national scale.

It is important to note that typically, all these impacts have an effect in a single production cycle. However, long-term impacts may also occur, including adjustments like behavioral changes in farmers that result in land use change (Biazin & Sterk, 2013; Henchiri et al., 2020). These long-term adjustments, while significant, are not measured or accounted for in this analysis.

Measurement of damages requires comparing actual conditions (hazard impact) as described above with counterfactual conditions (i.e. what would have happened in the absence of hazard). However, assessing the true counterfactual conditions is often challenging. A common practice in drought impact assessments in agriculture to compare agricultural production in drought years with that of recent non-drought years, which serve as a proxy for the counterfactual conditions. There is, however, a lack of consensus on the length of non-drought years, with some analyses using single-year (COPA-COGECA, 2003), three-year(Musolino et al., 2018), five-year(Trenczek et al., 2022), or six-year(Conradt et al., 2023) periods. Determining the optimal length of non-drought years to use as counterfactual conditions requires further research and is not addressed in this paper. Here, we use the five-year length to estimate counterfactual conditions following the approach of Trenczek et al (2022) as detailed in Sect. 2.2.

Another critical factor in defining counterfactual conditions is determining which years qualifies as a drought year. This becomes even more complicated due to the numerous factors influencing crop yields, such as soil quality, input materials, mechanization, and farm management practices, which can mask biophysical drought effects. Establishing indicators for drought declaration in the agricultural sector could prove useful in this regard. This would help consistently categorize a year and a region as drought or non-drought, ensuring accurate assessment of damages, even for small-scale drought events, and avoiding focusing solely on widespread droughts. We use an indicator based on the soil moisture, described in Sect. 2.4.

The empirical analysis is conducted in Germany, where the agricultural sector plays a significant role, with half of its land area dedicated to agricultural use (BMEL, 2022). The analysis is performed at the district level in Germany from 2016-2022, focusing on eight key field crops: winter wheat, winter barley, rapeseed, maize, spring barley, spring oats, sugar beets, and potatoes. Together, these crops account for 75% of Germany's agricultural area (Statistisches Bundesamt (Destatis), 2022a). Given that German agriculture is predominantly rainfed,  with less than 10% of the area equipped with irrigation (McNamara et al., 2024), our assessment primarily reflects damages on rainfed agriculture. The empirical analysis that follows focuses on the direct

biophysical impacts of these extremes and their role in farm revenues losses, excluding any indirect impacts beyond the immediate consequences of biophysically induced yield losses or the adaptation costs incurred by the farmers during droughts. Additionally, we assess the utility of high-resolution damage assessment, given that numerous studies suggest the need for such detailed assessment.

## 2.2 Damage Measurement

The damage $D$ in agricultural revenues during a drought year $t$ is quantified as the sum of difference between the expected revenue under counterfactual conditions and the actual revenue for each crop $c$ across eight crops. This can be expressed as:

$$D_t = \sum_{c=1}^{8} \left( \bar{R}_{expected,c,t} - R_{actual,c,t} \right) \tag{1}$$

where $\bar{R}_{expected,c,t}$ is the expected revenue for crop $c$, and $R_{actual,c,t}$ is the actual revenue for crop $c$ during the year $t$.

The counterfactual conditions aim to represent the average non-drought conditions specific to each region. In the context of ongoing climate variability, it is critical that the counterfactual conditions represent the evolving regional climatology (Suarez-Gutierrez et al., 2023) rather than relying on an idealized "normal" year in the traditional sense, which may no longer occur in practice. In this analysis, we define the counterfactual conditions as the average conditions in the preceding five non-drought years. We selected a five-year window following Trenczek et al. (2022), who used it to estimate damages for 2018 and 2019 droughts in Germany. The reason for this number of years is a trade-off: using more years could in theory further enhance the statistical representativeness regarding local climatic conditions, but it risks introducing bias by masking changing market and production conditions, as well as the overall trend in climate change, which also influence local yields and revenues (Lobell et al., 2011).

We determine drought (and non-drought) years based on the soil moisture. In order to do so, we use the Soil Moisture Index (SMI) metric, as explained in Sect. 2.4, and exclude any drought years in the average estimation, an improvement over existing approaches in the literature. This approach allows us to calculate revenue deviations using only normal (non-drought) years yield data without bias from multiple recent drought occurrences.

While the counterfactual is designed to exclude drought years, it is possible that some exposure to other extremes could still be reflected in the yields of non-drought years. Any potential yield anomalies in non-drought years, which could lead to over- or under-estimating drought damages, are addressed through the approach of estimating expected revenue based on the five-year average. The helps smooth out any random yield fluctuations and minimize the influence of non-drought related anomalies. Specifically, the expected revenue is estimated using the average yield over the preceding five non-drought years $i$ and the price in the drought year $t$, and actual revenue $R_{actual,c}$ is the revenue in the drought year. Therefore, for the present analysis, equation (1) can be rewritten as

$$D_t = \sum_{c=1}^{8} \left[ \left( \frac{1}{5} \sum_{i=1}^{5} Y_{i,c} \right) \cdot P_{t,c} - Y_{t,c} \cdot P_{t,c} \right] \tag{2}$$

where, $Y_{i,c}$ denotes the average crop yield for crop $c$ over the preceding five non-drought year $i$ (i.e., from year t−1 to t−5). $Y_{t,c}$ is the crop yield for crop $c$ in the drought year $t$, respectively, and $P_{t,c}$ is the price of crop $c$ in the drought year $t$. The use of drought-year prices to estimate expected revenues reflects contemporaneous market conditions during the drought year and maintains consistency with previous studies. While using in-year prices for estimating expected revenues might capture the indirect effects of droughts on prices (Badolo & Somlanare, 2012; Berhanu & Wolde, 2019; C. A. Ray, 2021), it would also incorporate other agricultural market developments unrelated to local droughts or extremes, complicating the attribution of damages to regional

extreme hydro-meteorological drivers. Holding prices constant ensures that the damage estimates focus solely on the yield changes induced by extreme hydro-meteorological drivers, providing a precise estimation of biophysically-induced direct damages in monetary terms.

To isolate the direct biophysical impacts of extreme hydro-meteorological drivers on crop yields from other influencing factors, we define the crop yield $Y_c$ for crop $c$ as a function of crop specific extreme-weather events (EWE), derived from data on precipitation (PR), temperature (T) and SMI:

$$Y_c = f_c(EWE_c) = f_c\big(g_c(PR_c, T_c, SMI_c)\big) \tag{3}$$

These crop yields are simulated using a statistical crop yield model, which is described in the next section. We use simulated crop yields to estimate actual revenue for drought years and expected revenue under counterfactual conditions for non-drought years, in order to calculate damages in eq.1. This ensures that the damage estimates are explicitly based on yield variability driven by EWE as described in equation 3, while excluding other factors unrelated to extreme hydro-meteorological drivers. Along with the assumption of constant prices, this methodology ensures that the revenue deviation between expected and actual revenues is attributed solely to the direct biophysically-induced yield impacts during droughts.

### 2.3 Statistical crop yield model

We apply a statistical crop yield model to isolate the impact of hydro-meteorological extremes including droughts on crop yields developed by Heilemann et al. (2024). The model predicts changes in crop yields based on different hydro-meteorological extremes, including drought. The statistical model is based on the Least Absolute Shrinkage and Selection Operator (LASSO) approach. It is a method for selecting relevant features via penalized multiple linear regression to avoid multicollinearity and obtain a higher predictive performance (Tibshirani, 1996). The statistical relationship between district-level crop yields and hydro-meteorological extreme variables was formulated using the following equation

$$Y = \sum_{j=1}^{p} \beta_j X_{ij} + \epsilon \tag{4}$$

Where $Y$ is the yield anomaly of a crop, $X_{ij}$ represents the vector of different crop-specific extreme weather events during sensitive growth phases in different months/seasons (explained below) and $\beta_1, \dots, \beta_p$ represent the model coefficients to be estimated. Each field crop used for the analysis is modeled separately.

By including the penalty parameter λ, the LASSO coefficients $\hat{\beta}_\lambda^L$ minimize the residual sum of squares of the regression models (James et al., 2013):

$$\sum_{i=1}^{n} \left( y_i - \sum_{j=1}^{p} \beta_j x_{ij} \right)^2 + \lambda \sum_{j=1}^{p} |\beta_j| \tag{5}$$

The model employs a 10-fold cross-validation to determine two key values of λ: $\lambda_{min}$, which minimizes the mean squared error (MSE) of the model, and $\lambda_{1SE}$, which is defined as $\lambda_{min}$ plus the standard error of λ that results in the minimum loss. Following the approach outlined by (J. Vogel et al., 2021), the stronger penalty term $\lambda_{1SE}$ is selected as a target, leading to the elimination of a greater number of variables compared to $\lambda_{min}$.

While we want to assess the damage of droughts on agriculture, other extreme weather events can co-occur and interact with drought. The statistical crop yield model employed accounts for this by taking 9 different extreme weather events into consideration (Table 1), which pose significant threats to crops in Germany, such as frost, heat, heavy rain, rain during harvest, precipitation

scarcity, drought, and waterlogging. By focusing on extreme events rather than mean temperature changes, the statistical yield model can more accurately capture the effects of extreme weather events (Webber et al., 2020), making it better suited for assessing the impact of such events (Newman & Noy, 2023). In Sect. 2.4, we describe how we delineate a drought occurrence and then estimate the compound effect of multiple weather extremes during the drought.

The timing of these events is crucial in determining crop damage. Therefore, the indicators for frost, heat, heavy rain, rain during harvest, and precipitation scarcity are included in the model as monthly features assessed during the relevant months of the growing season using crop-specific thresholds (Gömann et al., 2015). These indicators are calculated by counting the days in a month that exceed or fall below the defined thresholds. The indicators of drought and waterlogging are determined using the seasonal SMI value calculated from the monthly SMI value for the topsoil (25 cm soil depth), tailored to the growing period of each crop. To

this end, the monthly drought and waterlogging intensity as the difference between a SMI below 0.2 for drought, or above 0.8 for waterlogging is calculated. The model uses the seasonal drought and waterlogging intensity as the average of the monthly intensities. All features are used as continuous variables to account for stronger effects on crop yields through more intense extremes.

**Table 1Thresholds for extreme weather events from Heilemann et al. (2024)**

|  | Thresholds for extreme weather events | Time horizon of feature variable | Variable name |
|---|---|---|---|
| **Black frost** | Tmin < -25 / -20 / - 10 / - 5 °C | monthly | BF |
| **Late frost** | Tmin < 0 °C | monthly | LF |
| **Alternating frost** | Tmin > -3 °C & Tmax > 3 °C | monthly | AF |
| **Heat** | Tmax > 28 / 30 °C | monthly | Heat |
| **Heavy rain** | P > 20 mm/d | monthly | HR |
| **Rain during harvest** | P > 5 mm/d | monthly | RdH |
| **Precipitation scarcity** | P = 0 mm/d | monthly | PS |
| **Drought** | SMI < 0.2 | seasonal | Dr |
| **Waterlogging** | SMI > 0.8 | seasonal | Wl |


Based on the extreme event features, the LASSO models predict the annual yield anomaly (in %) as the dependent variable, representing the deviation of yields from the district-level mean yield for 1999-2022. Details on the standardized coefficients of the crop-specific LASSO models can be found in Table S2 of Heilemann et al. (2024). To illustrate the adequacy of the 1999–2022 period in identifying extremes, temporal histograms of all extreme weather events for the maize crop, used as a representative crop, are provided in the supplementary material (**Supplementary Figures S2-S3**). These histograms demonstrate that the selected

period captures a substantial number of extreme events, notably the exceptional droughts of 2003, 2018-2020, and 2022, waterlogging in 2001, 2007, 2010 and 2013 as well as severe frost and heat events. To simulate crop yields (in decitons per hectare - dt/ha), we multiply the predicted yield anomaly by the district-level mean yield. This approach allows us to isolate crop yields attributable to hydro-meteorological extremes defined in Table 1, including droughts. These simulated yields are then used for

damage assessment in drought-affected regions categorised using the SMI (as described in next section), aligning with the objective of quantifying the economic damages during droughts driven by the biophysical impacts of droughts and their interaction with other extremes. Descriptive statistics for the simulated yields, including their annual mean, minimum, and maximum values, are provided in Appendix A.

## 2.4 Drought categorization

To identify districts experiencing agricultural drought, we categorize the occurrence of drought in each district and year using the SMI (Samaniego et al., 2013) estimated from monthly soil moisture derived from the mHM. The $SMI_k$ represents the monthly soil water quantile at a grid cell at time $k$, relative to the range of historical observations. A given cell is considered experiencing a soil moisture drought when $SMI_k < \tau$. The threshold $\tau$ denotes that the cell is experiencing a soil moisture deficit occurring less than $\tau \times 100\%$ of the time. For our analysis, $\tau$ was set as 0.2 indicating moderate drought conditions that may pose potential harm

to crops and pastures (Zink et al., 2016). To consider the seasonal variations in water-supply-related impacts, we focus on the SMI during the active vegetative period from April to October. While recent studies have shown varying relationships between monthly SMI and crop yields (Peichl et al., 2021, 2021), we chose to utilize the average SMI during the active vegetative period to establish a neutral classification of drought impacting different crops.

Using monthly SMI data, at a resolution of 4km x 4km and covering the Germany entirely, the monthly average area under drought

conditions was estimated (Nagpal et al., 2024) for each district. The drought categorization based on the SMI reflects regional differences in climatic conditions as the SMI is calculated relative to the local historical soil moisture distribution in each district. To classify the occurrence of drought at a district level, it was considered that at least 20% area of each district must have an SMI<0.2 per month, and this condition should persist for at least three months during the active vegetative period i.e., the months of April to October in a given year (Belleza et al., 2023). This approach accounts for the slow development and spatial and temporal

accumulation characteristics of droughts. By using a threshold of SMI<0.2, we comprehensively capture all regions affected by droughts, including those experiencing varying intensities from severe (SMI<0.1) to exceptional conditions (SMI<0.02). This method enables the identification of non-drought years of a region, necessary for estimating expected revenues under counterfactual conditions. To evaluate the effect of this drought classification approach on damage estimates, we conducted sensitivity analyses by varying the threshold for the proportion of affected area ($\pm 5\%$), to confirm the robustness of damage

estimates under alternative drought classification criteria.

## 2.5 Data

### 2.5.1 Yield model inputs

Here, we provide a concise overview of the data used in the yield model used to analyze the direct biophysically-induced damage during drought on agriculture. Crop yields are simulated at the district level in Germany for eight field crops: winter wheat, winter

barley, rapeseed, maize, spring barley, spring oats, sugar beets, and potatoes, using the LASSO model. Detailed information on the input data used for yield estimation can be found in (Heilemann et al., 2024).

The annual yield data, used to simulate the yields, is sourced from the Federal Statistical Office of Germany available for the district level from 1999-2022 (Statistisches Bundesamt (Destatis), 2022b). Meteorological data encompassing minimum and maximum daily temperature and daily precipitation is obtained from the German Weather Service (DWD) through a network of

stations (*Deutscher Wetterdienst*, 2024). Additionally, the monthly SMI for Germany is derived from the mHM model (Samaniego et al., 2010, 2013).

### 2.5.2 Damage assessment

For the assessment of biophysically induced damages of extremes under droughts, we use data on crop acreage at the district level for the years 2016-2022. The data for cultivation on the arable land by crop (in ha) at the district level is collected periodically by

the statistical office in Germany and is not available for all years. Consequently, we use official statistical data for the years 2016

and 2020 (Statistisches Bundesamt (Destatis), 2020). For the remaining years, we rely on spatially explicit, remote-sensing-based crop maps with 10 m resolution for Germany (Blickensdörfer et al., 2022). The area under the eight crops analyzed in this study is extracted from the high-resolution crop map data at the district level using QGIS and R.

Yearly producer prices (€/dt) for crops in Germany are accessible from the European Statistical Office, except for sugar beets and maize (EUROSTAT, 2022). To achieve spatially-differentiated prices at a higher resolution, we scale this data using prices provided by the *Kuratorium für Technik und Bauwesen in der Landwirtschaft* (KTBL) calculator on the standard gross margin (KTBL, 2023b). For further details, please refer to Nagpal et al. (2024). For sugar beets, prices from KTBL at the country level are used, which were homogeneous until 2017 due to production limits imposed by the European Union and price guarantees provided to producers (Wimmer & Sauer, 2020). Since silage maize in Germany is not directly marketed but is used for fodder or biogas production (*FNR*, 2023), prices for silage maize are estimated by accounting for both these uses separately as described in Nagpal et al. (2024).

### 3 Results

### 3.1 Relevance of spatially disaggregated damage assessment

To show the utility of spatially disaggregated damage assessment and to understand the potential biases in using national-level data, we apply the methodology outlined in (Trenczek et al., 2022) using both national-level and regional-level reported crop yields, prices and land use data for Germany. The referenced report calculated damage estimates for 2018 and 2019 based on national-level reported data by determining the difference between expected and actual revenue. Expected revenue was derived from the average crop yields of the five year period of 2013-2017, combined with the prices and cultivated area from the assessment year.

While the report provided crop-wise damages specifically for winter wheat and silage maize and aggregate the damages for all other crops into a single category, our analysis extends this methodology to estimate damages for six additional crops: winter barley, rapeseed, spring barley, spring oats, sugar beets, and potatoes. In our analysis, crop-specific damages are calculated both at the national level, using aggregated national data, and at the regional-level, using reported yields from each district. Regional-level damages are then summed to obtain national totals for comparison with aggregated national-level results. This approach allows us to compare the extent of differences in damage estimates between national-level and regional-level data sources while retaining a crop-specific focus in both cases, providing insights into the potential biases that may arise from relying solely on national-level data.

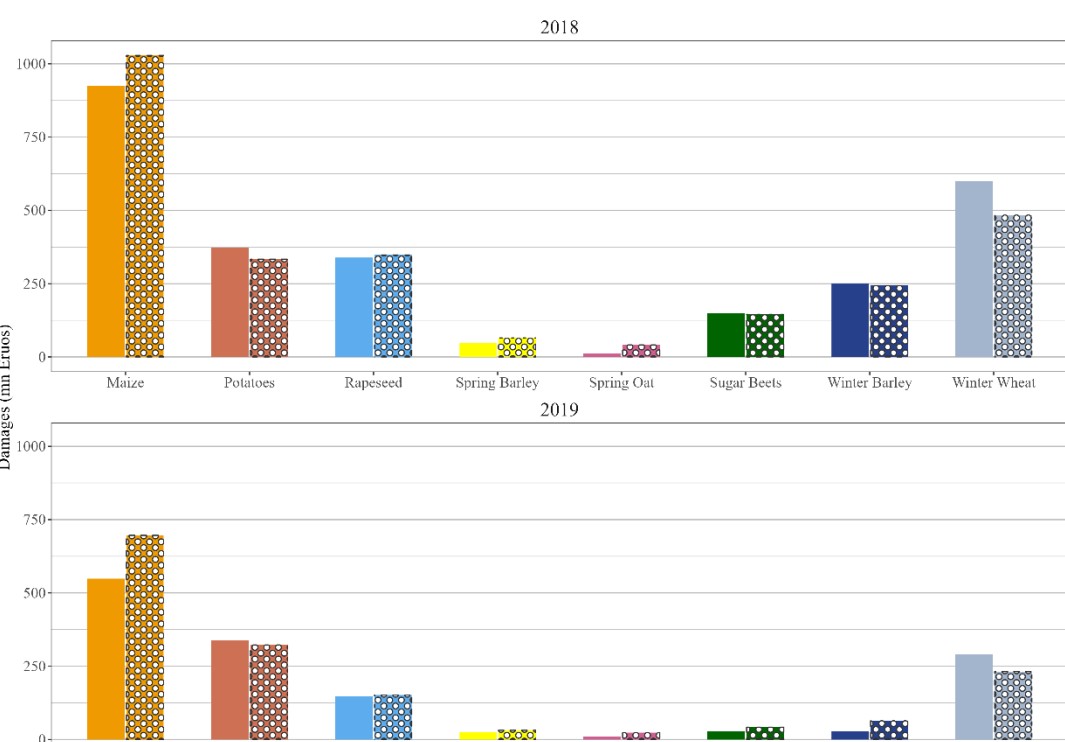

**Figure 2 Authors' crop-wise damage assessment based on the methodology outlined in (Trenczek et al., 2022) for the years 2018 and 2019 with both national-level and regional-level reported yield data for Germany.**

In our analysis, we found moderate difference between the total damages derived from national-level data and regional-level data. For 2018, the aggregated damages across all crops based on both national-level data and regional-level data are estimated at approximately €2.6 billion. For 2019, the aggregated damages across all crops based on national-level data (€1.4 billion) are slightly lower than those based on regional-level data (1.6 billion). However, there are notable differences in the damages across two major crops grown across Germany- maize and winter wheat (**Figure 2**). In both 2018 and 2019, the spatially distributed damages on winter wheat are lower than those based on aggregated national data, while they are significantly higher for maize. These results demonstrate that the use of spatially disaggregated data provides a more accurate quantification of crop-wise damages, which might not be captured by national-level assessments.

**3.2 Spatiotemporal analysis of direct biophysical damages**

Using the yields simulated by the statistical yield model (equation 4), we evaluate the direct biophysically-induced damages during droughts at the district-level in Germany from 2016 to 2022. This evaluation is done by comparing the actual revenue during a drought year with the expected revenue of non-drought years (equation 2). The revenues are estimated using simulated yields that isolate the direct biophysical impacts of extremes on crop yields from other influencing factors. The top row of panels in **Figure 3,** labelled as 'biophysical', shows the spatial distribution of these estimated biophysically-induced damages during droughts from 2016-2022.

Our analysis reveals that the average annual direct biophysically-induced damage across Germany, weighted by the proportion of agricultural area affected by drought (supplementary results 1), is estimated to be 781 million euros. The highest direct biophysically-induced damage occurred in the years 2018 and 2022, with revenue losses estimated at €1.7 billion and €850 million,

respectively. In northern Germany, a particularly notable decrease in revenues is observed, likely due to the substantial yield losses in these regions (**Supplementary Figure S6**).

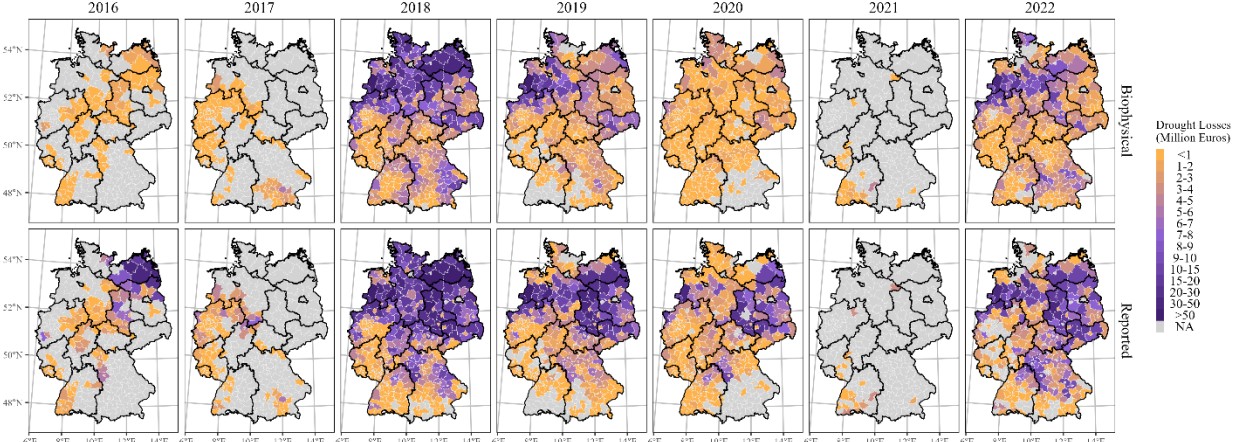

**Figure 3 Spatial distribution of the estimated total revenue losses during droughts in German district-level administrative units based on (top panels) yields simulated using statistical crop yield model that isolates the effect of hydro-meteorological extremes on yields and (bottom panels) reported yields reported in official statistics. The different colors indicate the total revenue losses (million Euros) in the districts.**

To further understand the relevance of impacts of extreme weather on agriculture during droughts, we compare the estimated direct biophysically-induced damages (using simulated yields) with the damages calculated from the yields reported in official statistics (hereafter called *reported* damages). This comparison helps understand the extent of direct damage specifically caused by extreme hydro-meteorological drivers on agriculture during droughts. The reported damages are presented in the bottom row of panels in **Figure 3**.

According to our analysis, the direct biophysically-induced damages account for an average of 45% of reported revenue losses during droughts between 2016 and 2022. In years with widespread droughts (2018, 2019, and 2022), the direct biophysically-induced damages represent an average of 60% of reported revenue damages (64%, 52%, and 65% respectively). These results demonstrate that the direct biophysically-induced damages of extremes constitute a considerable contribution to the overall revenue losses experienced by farmers during the period of widespread droughts in Germany.

### 3.3 Crop-wise analysis of direct biophysical damages

We present the aggregated crop-wise damages during droughts for four years with the highest revenue losses in Germany (2018, 2019, 2020, and 2022) in **Figure 4**. Our analysis reveals that silage maize suffered the most notable direct biophysically-induced damage due to droughts, followed by potatoes and winter wheat. When comparing these direct biophysically-induced damages with reported damages, we note a similar trend for maize and potatoes; however, reported losses for winter wheat are considerably higher than their direct biophysically-induced losses. Specifically, the impacts of extreme hydro-meteorological drivers on wheat crops are found to be 62% lower than the reported drought impacts. The situation is somewhat similar for other winter crops like winter barley and rapeseed. These findings indicate that drought-prone summer-grown maize and potatoes incur greater direct biophysically-induced damage compared to winter-grown wheat and barley. For maize, the direct biophysically-induced damage explain upto 97% (2018) of revenue losses and for winter wheat, upto 32% (2019). For the year 2020, the direct biophysically-induced damage of drought is significantly lower in comparison to the reported damages. This could be attributed to the fact that

the dry conditions in 2020 were primarily limited to the spring season (van der Wiel et al., 2023) and, therefore, had limited impact on crop yields (**Supplementary Figure S5**).

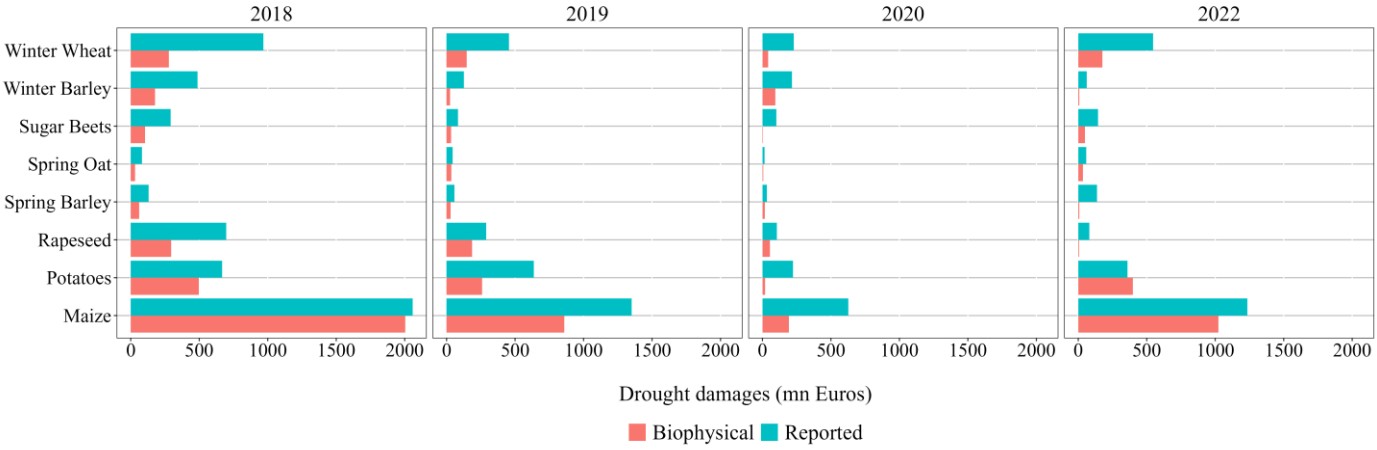

Drought damages (mn Euros)

■ Biophysical ■ Reported

**Figure 4 Crop-wise estimates of revenue loss in the four years with the largest aggregate losses during droughts across Germany based on yields simulated using the statistical yield model that isolates the impact of hydro-meteorological extremes on yields (orange bars, labelled 'biophysical') and yields reported in regional statistics (blue bars, labelled 'reported')**

The spatial distribution of direct biophysically-induced damages by crop for the four years with the highest revenue losses is depicted in **Figure 5**. The drought resulted in widespread revenue loss for almost all crops in Germany in 2018, 2019, and 2022 with some exceptions (like rapeseed in 2019 and 2022, spring barley and spring oats in 2022). Notably, potatoes experienced the highest revenue losses per ha amongst all crops across almost all districts in Germany given their high economic value (high yields per ha and high prices per ha). Drought-prone maize suffered significantly higher losses in the major production regions of the north (Lower Saxony and the surrounding districts) compared to the south (districts in Bavaria and Baden-Württemberg). In contrast, despite being the most widely cultivated crop across Germany, winter wheat showed much lower revenue losses than maize. In 2020, spring barley incurred more widespread crop losses than any other crop. Interestingly, in 2019, 2020, and 2022, only limited losses were observed for sugar beets in Mecklenburg-Vorpommern and the bordering districts of Lower Saxony and Saxony-Anhalt, despite a considerable share of area in these regions dedicated to growing this crop.

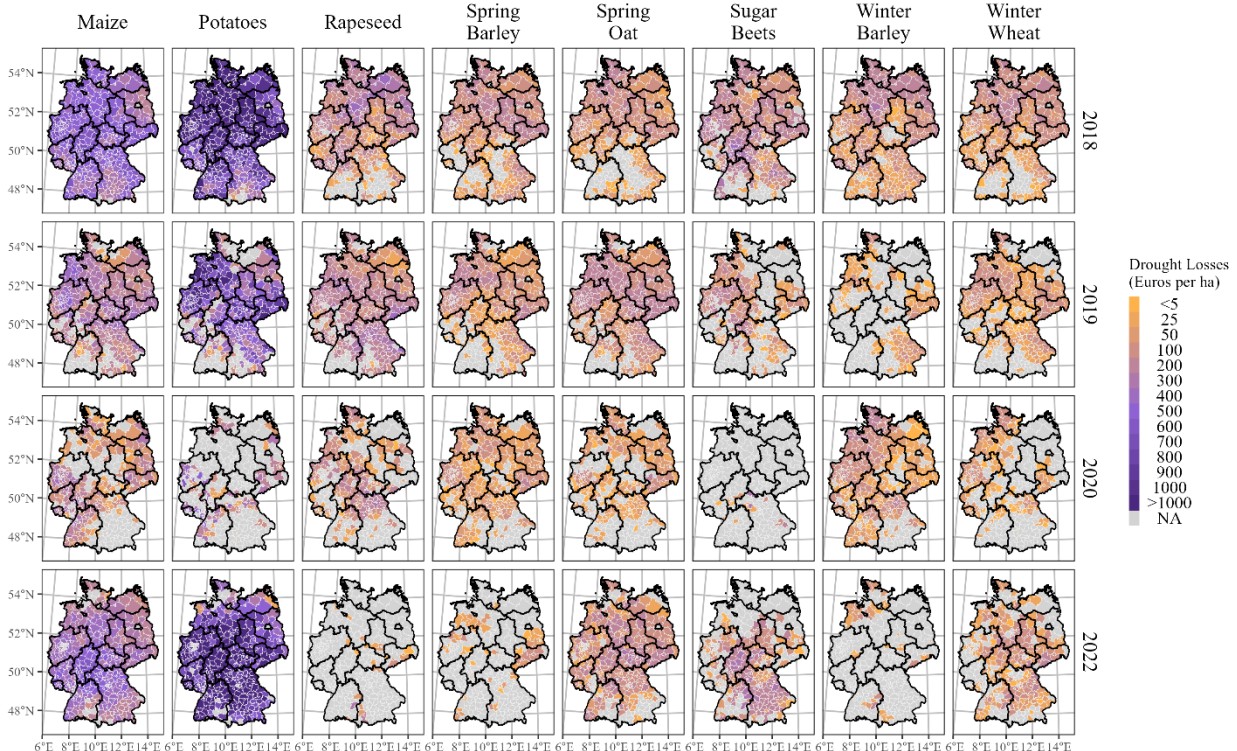

**Figure 5** Spatial distribution of direct biophysically-induced crop specific damages during droughts in German district-level administrative units in the four years with the highest revenue losses. The different colors indicate different levels of revenue losses (in Euros per ha) in the districts.

### 3.4 Contribution of droughts and various hydro-meteorological extremes to direct biophysically-induced damages

Next, we examine to which degree droughts and other hydro-meteorological extreme events contributed to fluctuations in yields during 2016-2022, in order to understand the relative importance of their impacts on agriculture. This is done by calculating the feature contributions to the predicted yield change using the coefficients estimated with the LASSO models at the optimal penalty parameter $\lambda_{1SE}$ (Heilemann et al., 2024). **Figure 6** displays the average contribution of various hydro-meteorological extremes to yield anomalies across Germany, which vary by crop and year. Contrary to intuition, some extremes also have positive effects on yield anomalies, although this is dependent upon the season/month of occurrence and the intensity of extremes, and the specific crop affected (Heilemann et al., 2024; Schmitt et al., 2022).

In 2016, 2017, and 2021, positive yield effects from weather extremes outweighed the negative impacts on crop yields. Despite limited drought-affected areas in Germany (**Supplementary Figure S4**), the negative impacts of droughts are evident in various crops during these years. Except 2020, the years with widespread droughts in Germany (2018, 2019, and 2022) saw droughts and heat contributing to negative yield anomalies for almost all crops. While there are some exceptions (sugar beets in 2018, and spring oats in 2019), droughts generally cause more severe impacts than heat. In 2019, the effect of drought, and heat was coupled with precipitation scarcity during spring (meteorological drought) which led to notable negative yield anomalies in spring oats and, to some extent, in spring barley and winter wheats. In contrast, negative yield anomalies in 2020 were largely driven by meteorological drought during spring instead of soil moisture drought. Meteorological droughts during spring commonly threaten agricultural productivity, as sufficient rainfall in spring is critical for distributing fertilizers throughout the soil (Gömann et al., 2015). These results show the complex interplay of weather extremes and their varying combinations, which determine the extent of yield losses in different years.

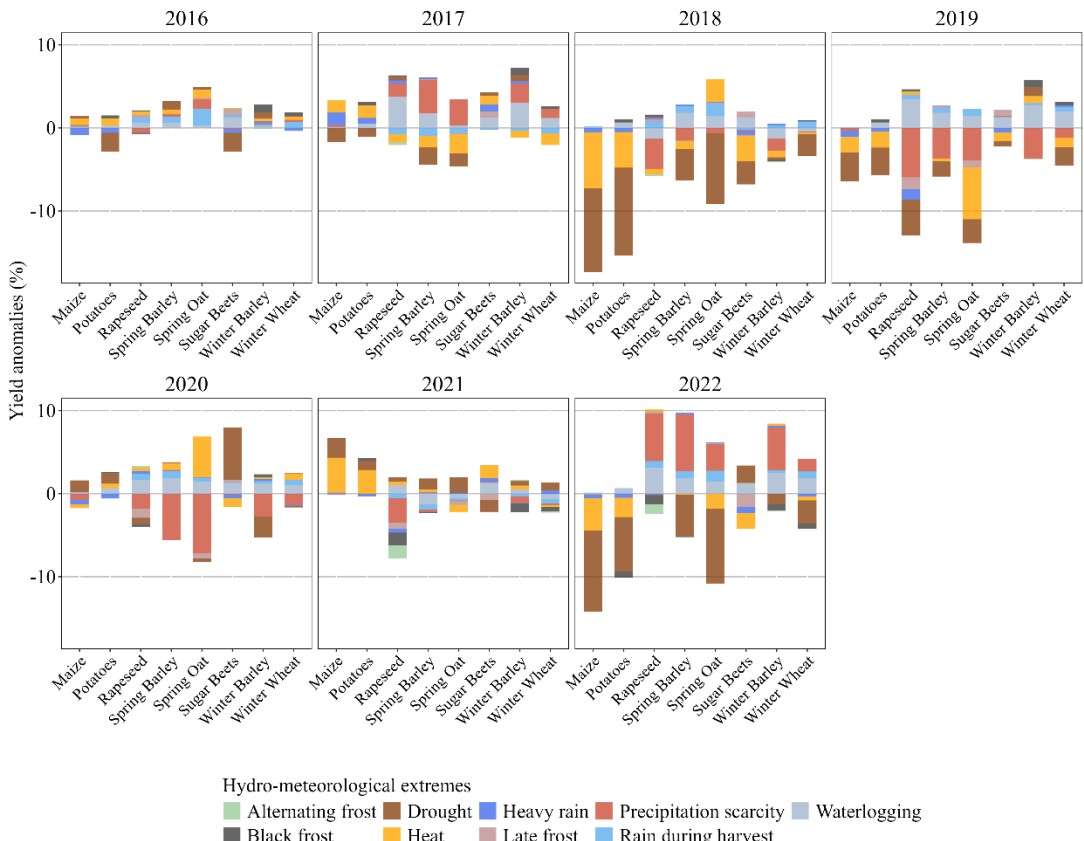

Figure 6 Contribution factors of hydro-meteorologically extremes to yield anomalies across different crops computed from the LASSO regression model.

### 3.5 Sensitivity analysis of estimated direct biophysically-induced damages

To evaluate the robustness of the direct biophysically-induced damages, two sensitivity analyses were conducted: (a) adjusting the counterfactual period to include 4-year and 6-year averages for estimating expected revenues, and (2) modifying the drought classification by testing variations in the area threshold of each district with an SMI < 0.2 per month. Specifically, we tested ±5% changes in the original 20% threshold for the categorisation of district affected by drought.

In our analysis, we found that the average annual damage estimates ranged from €766 million (under the 6-year counterfactual) to
€812 million (under the 4-year counterfactual). Adjusting the drought area threshold led to variations in damage estimates, ranging from €767 million (under a 5% decrease in the threshold) to €798 million (under a 5% increase in the threshold). The main results presented in previous sections fall within the range of these variations, underscoring the robustness of the estimates while accounting for potential uncertainties in the counterfactual definition and drought classification area.

**Figure 6** presents the annual direct biophysically-induced damages for years 2016-2022 across all sensitivity scenarios. The figure
shows that the temporal patterns are consistent across scenarios.

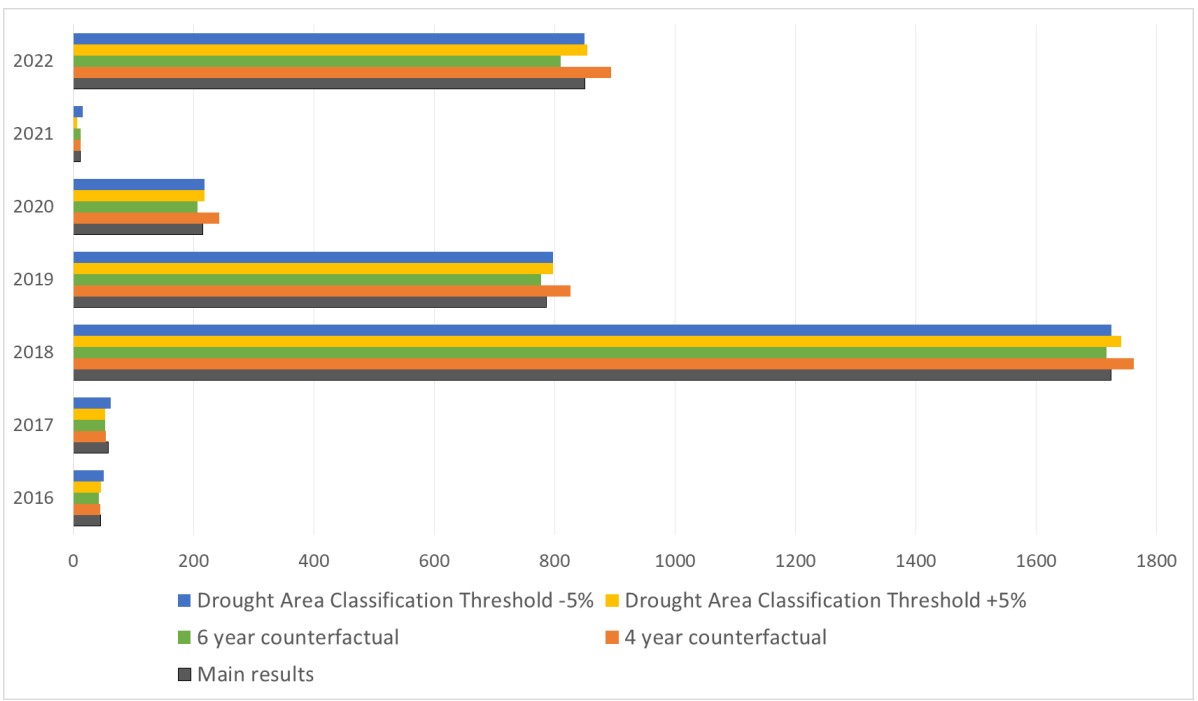

**4 Discussion**

For our analysis of direct biophysically-induced damages of extremes during droughts, we aggregate the impacts of eight field
crops in Germany. The direct biophysically-induced damages of droughts were estimated by comparing the revenue generated
during the drought year with that of the preceding five non-drought years across all districts in Germany. Recent research by Di
Marcoberardino & Cucculelli (2024) has highlighted the significant impact of extreme events like droughts and heatwaves on the
local economies across Europe, underscoring their localized nature. Providing a spatially distributed assessment is especially
important for enhancing risk management, as it can help communicate risk to stakeholders and inform targeted policies and support
programs (Brás et al., 2021; Rose, 2004). Our analysis comparing regional-level and national-level data for estimating drought
damages reveals that using spatially disaggregated information yields more accurate assessments of revenue losses by crop, which
may not be reflected in national-level assessments. The spatially distributed approach used here can be adapted in other regions to
provide more precise assessment of revenue losses and to inform policy planning.

Our findings reveal that the average direct biophysically-induced damage driven by extremes during droughts from 2016 to 2022
was €781 million per year (sensitivity range: €766 million-€812 million) across Germany. The years 2018 and 2022 experienced
the highest losses, estimated at €1.7 billion and €850 million respectively. The spatial distribution of the total damage we found
for 2018 is consistent with previous research. The sector-wise analysis of the impacts of droughts for 2018, conducted by de Brito
et al., (2020) showed that agriculture in eastern Germany had the highest impacts. Conradt et al. (2023) found that the German part
of the Elbe River basin in northern Germany suffered the highest yield losses in 2018. During years of widespread droughts, the
revenue losses were greater in northern Germany compared to southern Germany. In southern Germany, there is some evidence
that drought stress has little impact on crop yields (Lüttger & Feike, 2018). Our analysis of the spatial distribution of annual average
yield loss for all crops during droughts across Germany also found similar patterns (Supplementary Figure S6). These findings
underscore the need for spatially targeted polices and interventions, particularly in northern and eastern Germany, where agriculture
is disproportionally affected during droughts.

The comparison of the direct biophysically-induced revenue losses with reported losses shows that in years of widespread drought, biophysical factors like hydro-meteorological extremes explain 60% of the revenue losses in Germany. These losses are largely driven by varying combinations of droughts, heat, and precipitation scarcity. This is consistent with emerging research on the joint impacts of extreme events on crop yields, which has identified drought and heat as the most relevant concurrent extremes in Europe, both in the current and future climate (Brás et al., 2021; Orth et al., 2022; von Buttlar et al., 2018; Webber et al., 2018). The

contribution of this study lies in quantifying the extent to which economic damages are directly driven by the biophysical yield impacts of these drivers. It helps disentangling the contributions of extreme hydro-meteorological drivers of yields vis-à-vis other drivers of yields to revenue losses, underlining the importance of these factors in shaping agricultural outcomes. While several weather extremes driving damages during droughts have been assessed and included, this assessment cannot be considered comprehensive. Important factors such as the impacts of pests and diseases (Khodaverdi et al., 2016; Meisner & de Boer, 2018),

soil water retention capacity (Blanchy et al., 2023), as well as farm management practices (Soares et al., 2023) are not included in these damage estimates.

The crop-wise examination of revenue losses during drought in Germany revealed that summer crops like maize suffered the highest aggregate losses, followed by potatoes. The maize crop is particularly vulnerable to droughts, as highlighted by previous studies (Schmitt et al., 2022; Webber et al., 2020) and this vulnerability was evident in the high revenue losses we observed in

almost all years. According to our analysis, up to 97.4% (2018) of maize's revenue loss can be explained by the direct biophysical impacts of extremes. These results are consistent with findings of Reinermann et al. (2019) who analyzed drought impacts using satellite-based vegetation indices. Interestingly, potatoes, which are typically considered a high-value cash crop grown under irrigation, suffered the highest losses in Lower Saxony, a region with extensive irrigation infrastructure. This could be because the potato yield losses during droughts are mostly due to increased temperatures, rather than a reduction in precipitation which could

be mitigated through irrigation only upto a certain degree (Egerer et al., 2023).

In comparison to our findings, García-León et al. (2021) estimated that agricultural losses due to droughts in Italy ranged from €0.55 billion and €1.75 billion per year, while Howitt et al. (2015) reported crop revenue losses in California, United States of approximately $902 million to $940 million per year. Our result that maize was the most effected crop during recent droughts in Germany is consistent with the findings of Brás et al (2021), who found maize as experiencing the highest production losses among

cereals across Europe due to droughts and heatwaves between 1964 and 2015. Maize's vulnerability to drought is not limited to Europe. In the United States, substantial yield variability in maize has been linked to drought and heat stress (Zipper et al., 2016). Similarly, in China, maize yield losses have been shown to increase with the severity of drought, contributing to significant reductions in maize production across the country (S. Liu et al., 2022). These comparisons highlight the dual challenge of mitigating economic losses across diverse cropping systems and addressing the specific vulnerabilities of drought-sensitive crops like maize.

They underscore the importance of globally coordinated efforts to enhance agricultural resilience in the face of increasing weather extremes.

While our estimates provide robust insights into the biophysical damages of droughts and associated extremes in drought affected regions, there are some limitations to consider. First, our analysis is focused on short-term impacts damages and does not include adaptation costs or indirect impacts beyond the immediate consequences of biophysically induced yield losses. Second, the

estimation of revenue losses might be underestimated due to the inherent limitations of the statistical yield model in simulating extreme crop yields. This underestimation partially arises from the use of pre-defined thresholds for extreme events. Since the study relied on an established statistical model, we did not assess the sensitivity of these thresholds, which should be explored in future research to improve robustness. Last, this yield model is based on anomalies relative to district-level means which limits our ability to fully control for the biophysical impacts of weather extremes in the counterfactual. While a non-extreme weather

events counterfactual could have provided valuable insights into the interplay between droughts and other extremes, this was not feasible within the current modelling framework. Future research should focus on testing different types of yield models that allows control of impacts of weather extremes in the counterfactual while capturing the dynamics of extreme weather impacts on yields.

**5 Conclusion**

This study presents a conceptual framework to facilitate the understanding and estimation of economic impacts of hydro-
meteorological extremes associated with droughts in agriculture. Within the framework, we measured spatially distributed, direct biophysically-induced damages on farmers' revenue at the district level in Germany during droughts. Our estimates bridge gaps related to consistent economic damage assessment that can be used for the assessment of the costs of climate change (Frame et al., 2020). Farmers' decision-making in the context of drought would also benefit from such analysis, especially if these assessments are extended and linked with drought monitoring and early warning systems (Muller et al., 2024). Additionally, we show the utility
of spatially distributed data for accurate crop-specific damage assessments.

Our analysis revealed an average annual revenue loss due to biophysical impacts of extremes of €781 million across Germany during drought, accounting for 45% of reported revenue losses. In years with widespread droughts (2018, 2019, and 2022), the direct biophysically-induced damages represent an average of 60% of reported revenue loss, highlighting the dominant role of hydro-meteorological extremes in driving the revenue losses experienced by farmers. By isolating the impacts of hydro-
metrological extremes from other drivers of farm revenue losses in droughts, the findings emphasize the critical need to adapt to such extremes not only in the present-day climate but also in the future, where such extremes are expected to become more frequent and intense.

Our results underscore the role of hydro-metrological extremes in revenue losses during droughts in Germany. Specifically, for drought-prone, summer-grown crops like maize, the hydro-meteorological extremes, such as reduced soil moisture, can explain
upto 97% of the reported losses in 2018. In contrast, for the winter-grown crops like wheat, the contribution of hydro-meteorological extremes is less pronounced, explaining upto 32% of the reported losses in 2019. These results can guide more targeted adaptation during droughts, focusing on specific crop types. For example, insuring summer-grown crops against simultaneous or successive extremes, such as drought and heat, or enhancing breeding effectiveness.

While this study provides detailed understanding of biophysical damages during droughts, future research could expand the
analysis to include adaptation costs, indirect impacts, and a more refined counterfactual approach to better capture the interplay between weather extremes. Nonetheless, our analysis provides valuable insights into the far-reaching economic consequences of droughts in the agricultural sector. These insights should be of significant interest to decision-makers, guiding the development of effective strategies for mitigating the effects of droughts and implementing measures to build resilience in affected regions. Future work should focus on routinely estimating these losses within operational drought monitoring systems such as the German Drought
Monitor (Zink et al., 2016), and forecasting frameworks like Hydroclimatic Subseasonal-to-Seasonal forecasting system (*Hydroclimatic Forecasting System*, 2024). By linking hydro-meteorological variables with projected economic damages, such integration would enhance early warning capabilities, improve economic preparedness against increasing weather extremes, and support more proactive adaptation strategies.

**Appendix A: Descriptive statistics for simulated crop yields, including annual mean, minimum, and maximum values for**
**each crop during the 2016-2022**

| Crop/Year | Mean Yield (dt/ha) | Min Yield (dt/ha) | Max Yield (dt/ha) |
|---|---|---|---|

**Maize**

| | | | |
|---|---|---|---|
| **2016** | 456.88 | 212.62 | 599.47 |
| **2017** | 459.23 | 224.28 | 596.93 |
| **2018** | 369.89 | 182.05 | 488.38 |
| **2019** | 421.74 | 184.88 | 525.93 |
| **2020** | 444.48 | 207.60 | 551.97 |
| **2021** | 480.78 | 226.55 | 644.59 |
| **2022** | 383.01 | 194.34 | 503.36 |

**Potatoes**

| | | | |
|---|---|---|---|
| **2016** | 371.61 | 230.49 | 523.78 |
| **2017** | 381.81 | 240.83 | 563.83 |
| **2018** | 320.73 | 211.40 | 472.44 |
| **2019** | 356.77 | 212.72 | 519.60 |
| **2020** | 377.98 | 220.23 | 541.00 |
| **2021** | 388.03 | 246.48 | 572.54 |
| **2022** | 335.88 | 197.96 | 518.20 |

**Rapeseed**

| | | | |
|---|---|---|---|
| **2016** | 36.24 | 20.61 | 46.47 |
| **2017** | 36.80 | 18.21 | 47.82 |
| **2018** | 34.13 | 17.09 | 46.00 |
| **2019** | 32.80 | 17.44 | 42.66 |
| **2020** | 35.45 | 20.80 | 48.23 |
| **2021** | 35.33 | 21.47 | 44.95 |
| **2022** | 38.52 | 20.54 | 50.55 |

**Spring Barley**

| | | | |
|---|---|---|---|
| **2016** | 50.45 | 22.13 | 75.95 |
| **2017** | 50.11 | 21.06 | 71.83 |
| **2018** | 47.56 | 20.78 | 72.09 |
| **2019** | 47.86 | 18.73 | 71.69 |
| **2020** | 48.22 | 20.62 | 77.13 |
| **2021** | 49.30 | 22.77 | 73.02 |
| **2022** | 51.64 | 21.92 | 77.76 |

**Spring Oat**

| | | | |
|---|---|---|---|
| **2016** | 49.51 | 22.56 | 72.71 |
| **2017** | 47.03 | 21.59 | 66.36 |
| **2018** | 45.25 | 21.42 | 63.03 |
| **2019** | 42.13 | 14.64 | 64.48 |
| **2020** | 46.40 | 20.40 | 68.18 |
| **2021** | 46.16 | 22.01 | 67.33 |

| | | | |
|---|---|---|---|
| **2022** | 45.27 | 20.52 | 66.36 |
| **Sugar Beets** | | | |
| **2016** | 641.65 | 444.39 | 834.22 |
| **2017** | 660.26 | 459.06 | 844.77 |
| **2018** | 616.90 | 417.44 | 821.43 |
| **2019** | 642.56 | 444.91 | 841.17 |
| **2020** | 682.29 | 472.42 | 892.76 |
| **2021** | 655.47 | 455.41 | 858.88 |
| **2022** | 637.20 | 443.13 | 874.71 |
| **Winter Barley** | | | |
| **2016** | 66.53 | 38.09 | 88.69 |
| **2017** | 68.56 | 36.95 | 93.26 |
| **2018** | 62.45 | 32.75 | 85.36 |
| **2019** | 66.40 | 35.90 | 91.93 |
| **2020** | 63.04 | 36.30 | 86.66 |
| **2021** | 66.19 | 38.69 | 91.92 |
| **2022** | 68.81 | 37.26 | 93.76 |
| **Winter Wheat** | | | |
| **2016** | 73.64 | 39.60 | 96.00 |
| **2017** | 73.05 | 38.34 | 97.14 |
| **2018** | 70.92 | 36.29 | 94.15 |
| **2019** | 71.81 | 36.09 | 99.69 |
| **2020** | 73.17 | 39.35 | 96.68 |
| **2021** | 73.19 | 39.66 | 99.00 |
| **2022** | 72.75 | 37.92 | 99.85 |

**Author contribution**

MN, BK, EG, and CK conceptualized the study; MN, JH, and CK developed the methodology; MN and JH curated the data and developed the software; MN conducted the formal analysis, visualization and prepared the original draft of the manuscript. JH, LS, BK, EG, and CK reviewed and edited the manuscript. BK, EG, and CK supervised the research.

**Declaration of competing interests**

The authors do not have any competing interests.

**Acknowledgements**

We would like to thank Edward Digman for proofreading this article. The authors acknowledge the assistance of ChatGPT in enhancing the language quality. Following the use of this tool, the authors have carefully reviewed and edited the content as necessary, and they assume full accountability for the publication's content.

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
