# Peer review of "Measuring extremes-driven direct biophysical impacts in agricultural drought damages"

_EGUsphere, 2024_

## Referee Comment (RC2)

Review of "Measuring extremes-driven direct biophysical impacts in agricultural drought damage"

The authors address the complicated question of accurately estimating the direct impacts of droughts on agricultural yields. In doing so, they tackle a number of issues that confound the drought estimates, including the co-occurrence of other extreme weather events, the regional heterogeneity in occurrences and effects that limit the viability of national aggregated measures and the presence of indirect effects that come from secondary and tertiary impacts. Using Germany as the backdrop, they find that the direct impact of droughts amounts to 781 million euros in the period investigated, accounting for 60% of reported yield losses in drought years, going as far as 97% of total damage when the focus is on rice yields in 2018. They also find a discrepancy when comparing national aggregated estimates to regionally estimated losses, suggesting a preference for regional estimates.

Some issues remain and are addressed below

1. The first issue I came across while reading was confusion on what exactly was being investigated. For the first few pages, I assumed the purpose was an investigation of the impact of agricultural droughts measured by soil moisture, but after a few pages, the phrase "extreme weather on agriculture during drought years" gave the impression that the investigation was a secondary effect of other extreme weather events during drought years. After reading, I am convinced that the paper is just about the impact of drought (first, with a combination of other extremes investigated in section 4.4), if I am mistaken, it adds to the confusion I had while reading through. Simplifying the text and stating precisely what was investigated would be ideal.

2. The measure of damage in equation 1 itself may be over or underestimating drought effects in its current form. With the impact being the difference between the expected revenue and the actual revenue, it ascribes this difference in its entirety to drought effects, which may not be entirely true. It is the classic diff-in-diff argument. For the damage equation to be solely due to droughts, the authors current approach would necessitate that in non-drought years, expected outcomes ALWAYS match the realized outcomes. I am doubtful that this is true, and as such, any shortfalls in non-drought years would imply that negative drought effects are overestimated while any windfalls (realized yields greater than expected) would underestimate the drought effects. Therefore, I suggest that the damage be estimated as

$$D_t = \sum_{c=1}^{8} \left( \overline{R_{expected,c,t}} - R_{actual,c,t} \right) - \frac{1}{T} \sum_{t=1}^{T} \sum_{c=1}^{8} \left( \overline{R_{expected,c,t}} - R_{expected,c,t}^{ND} \right)$$

Where the additional term is the average difference between expected revenue and realized revenue in T non drought years in the study. This way, any non-drought related discrepancies can be correctly accounted for.

3. In equation 2, using the current price to estimate expected revenue might be problematic given that others have found that extreme weather events have their own distinct impact on prices (Berhanu & Wolde, 2019; Felix & Romuald, 2012; Ray, 2021). It may be beneficial to use in year prices adjusted for inflation to estimate expected revenues. If the idea was to allow for the focus to be just on yields, then I would recommend just leaving prices out entirely. Including prices would mean that expectations are driven by two sources: expected yields and expected prices, both of which can be separately impacted by domestic and external weather shocks.

4. The statistical crop yield model shows a regression that included several weather extremes on the right-hand side, but did not discuss how the drought contribution to yield was extracted or what it in fact looks like. Some descriptive statistics would be helpful here. Is drought driven yield just beta*drought? Is the dependent variable in subsequent analysis yields as a result of droughts? More exposition on what exactly was done to generate the variable of interest would be ideal.

5. The study simultaneously addresses two separate issues in its spatial disaggregation exercise. From my reading, the study disaggregates crops, as well as the country and it is not clear which of these is responsible for the differential when compared to national figures. This is especially true as the only differences come when crops are broken out and investigated individually. To summarize, would the national estimate lead to the same discrepancy without spatial disaggregation if the damage of each crop is investigated separately? (Basically, is the difference a result of disaggregating crops or spatial disaggregation)

Some typos…

Page 2 line 64: underestimates should be underestimate

Page 2 line 77 "…are derived from **a** the…" delete "a"

Page 3 line 97 "casual" should be "causal"

References

Berhanu, M., & Wolde, A. O. (2019). Review on climate change impacts and its adaptation strategies on food security in sub-Saharan Africa. *Agricultural Socio-Economics Journal*, 19(3), 145–154.

Felix, B., & Romuald, K. S. (2012). Rainfall shocks, food prices vulnerability and food security: Evidence for sub-saharan african countries.

Ray, C. A. (2021). The Impact of Climate Change on Africa's Economies.

---

## Referee Report (RR1)

Thank you for revising your manuscript. I've read the revised manuscript and your response letter again and am mostly satisfied with your responses. I do have two remaining comments and a handful of minor comments aimed at clarification.

Remaining comments:

1. My first comment relates to the unit of analysis (field, farm, region?) and the aim of the paper (lines 73-75). I still struggle to understand what the aim of the paper is, as it consists of two parts. The first part is about the biophysical damage of extremes and droughts, and the second part relates to farm-level revenue losses. I am a bit confused here, as you work with regional/district-level data and present your results (i.e. Figures 3-7) at a regional level as well. Can you clarify how you measure farm-level revenues with regional data? Or is the unit of analysis regional/district-level? Please check throughout your whole manuscript if it is regional.

2. My second comment relates to the focus on droughts or droughts and other extreme weather events in the current manuscript. Having read your work again, I have to admit that I still feel the manuscript predominantly focuses on droughts and less on other extremes. You now acknowledge this in the limitations (lines 500-505), where you highlight that these other extremes are not considered in the counterfactual.

   Besides that, the vast majority of the results focus on droughts. Section 3.4 should be about droughts and other extremes, but I find it still to be dominated by droughts, with little mention of other extremes. Can you describe this a bit better in the main text? Figure 6 is clear, so you could build on that.

A handful of minor comments, most of them for clarification:

- You often refer to "regions" (e.g. line 187) and sometimes to "districts" (e.g. lines 227-229). Can you define somewhere what you mean by these? Do regions consist of districts (i.e. a region is bigger than a district), or are they synonyms?
- Figures 3, 4, and 5: Check the legend and be consistent. Figures 3 and 5 refer to "drought losses," while Figure 4 refers to "drought damages." Should this be consistently "damage"?
- Figure 4: "mn" should be "millions."
- Lines 395-400: I would refrain from referring to specific districts without specifying where in Germany these districts are located. For non-German readers, that is hard to understand. Specify where Mecklenburg-Vorpommern, Lower Saxony, and Saxony-Anhalt are located.
- Lines 408-410: "Contrary to intuition,… specific crop affected." Explain what extremes have a positive effect on yield anomalies.
- Lines 430-435: Thanks for running all these robustness checks and sensitivity analyses. Can you add a couple of lines explaining these findings? What explains the lower and/or upper range? And how do these results increase your confidence in your main model specification?
- Lines 506-508: "This study presents a conceptual framework … in agriculture." You just removed the conceptual framework when you revised the paper. Maybe rephrase it to "provides an empirical illustration"? I would also change "economic impacts" to "economic damage" in that sentence to maintain consistency throughout the paper.
- You suggest future research both at the end of the discussion (lines 504-505) and at the end of the conclusions (lines 531-535). I suggest merging these. Either at the end of the discussion or the conclusion is fine.

---

## Author Response (AR3)

**Helmholtz Centre for Environmental Research – UFZ**
Permoserstr.15 · 04318 Leipzig · Germany

Dr. Giulia Vico
Editor
Natural Hazards and Earth System Sciences

**Mansi Nagpal**
PhD Student
Department of Economics
Fon +49 341 6025-4679
mansi.nagpal@ufz.de

Leipzig, 4/11/2025

**Cover letter for manuscript number EGUSPHERE-2024-2585**

Dear Dr. Vico,

We appreciate the continued consideration of our manuscript **"Measuring extremes-driven direct biophysical impacts in agricultural drought damages"**, and the thoughtful feedback provided by the reviewers during this second round of review.

In this revised version, we have addressed the remaining points raised by the reviewers. Specifically, we have clarified the unit of analysis, expanded the discussion of hydro-meteorological extremes beyond drought, and added new supporting material to better reflect the role of multiple extremes in our assessment (including the bar plot in Figure 3 and Supplementary Figure S7). A detailed, point-by-point response to all comments is included.

We sincerely appreciate the thoughtful and constructive feedback provided throughout the review process. We believe the manuscript has been now significantly improved and look forward to your further assessment.

Best regards,
Mansi Nagpal on behalf of all co-authors

**Helmholtz Centre for Environmental Research – UFZ**

Company domicile: Leipzig

Permoserstr. 15, 04318 Leipzig, Germany
or
PF 500136, 04301 Leipzig, Germany

info@ufz.de
www.ufz.de

Registration court: Leipzig district court
Commercial register No. B 4703

Chairman of the Supervisory Board:
MinDirig'in Oda Keppler

Scientific Director:
Prof. Dr. Rolf Altenburger

Administrative Director:
Dr. Sabine König

Bank details:
HypoVereinsbank Leipzig
Sort code 860 200 86
Account No. 5080 186 136
Swift (BIC) code HYVEDEMM495
IBAN No. DE12860200865080186136
VAT No. DE 141 507 065
Tax No. 232/124/00416

[Figure]

[Figure]

Manuscript Number: **egusphere-2024-2585**

**Response to Reviewers comments**

11 April 2025

**Contents**

**Reviewer comments and authors' replies marked in black**

**Previous manuscript and supplement text marked in brown**

**New manuscript and supplement edits marked in blue**

All page and line numbers refer to the revised, unmarked manuscript

**Response to Reviewer#1**

We sincerely thank the reviewer for the positive assessment and constructive feedback. We have carefully addressed the two remaining comments as well as the minor suggestions for clarification, and revised the manuscript accordingly. Our detailed responses to the remaining comments are provided below.

**Remaining comments**

| S.No. | Reviewer's Comment | Authors' Response |
|---|---|---|
| 1 | My first comment relates to the unit of analysis (field, farm, region?) and the aim of the paper (lines 73-75). I still struggle to understand what the aim of the paper is, as it consists of two parts. The first part is about the biophysical damage of extremes and droughts, and the second part relates to farm-level revenue losses. I am a bit confused here, as you work with regional/district-level data and present your results (i.e. Figures 3-7) at a regional level as well. Can you clarify how you measure farm-level revenues with regional data? Or is the unit of analysis regional/district-level? Please check throughout your whole manuscript if it is regional. | Agreed and clarified: We have removed the term "farm" from the aim of the paper to prevent confusion about the unit of analysis. Throughout the manuscript, we now clarify that our analysis is conducted at the district (regional) level. While the term 'farm-level' was based on our conceptualization of the revenue losses, we recognize that this might cause confusion and is not necessary for understanding the paper. All data inputs, simulations, and resulting figures (e.g., revenue losses) are based on district-level aggregations, and this has been made explicit in the revised manuscript. To further improve clarity, we have revised the text surrounding the study's aim in the introduction to explicitly describe the second aim of the paper.
 Pg 2, lines 73-82: *"The aim of this study is to measure the direct biophysical damage of extreme hydro-meteorological drivers during droughts (hereafter called direct biophysically-induced damages) and assess the contribution of these biophysically-induced damages to the total reported agricultural revenue losses. These damages refer to the loss in revenue caused by the effects of extreme hydro-meteorological drivers on crop yields, without accounting for other economic impacts, such as changes in costs. They include the effects of droughts themselves, as well as additional damage from concurrent or successive weather extremes that exacerbate drought-related effects in regions experiencing drought conditions. To isolate the biophysical impacts of these extremes on crop yields from other influencing factors, we employ crop specific statistical yield models. To evaluate the contribution of these extremes in shaping observed economic outcomes, we compare the direct biophysically-induced damages estimated* |

| S.No. | Reviewer's Comment | Authors' Response |
|-------|--------------------|--------------------|
| | | *from these models with reported revenue losses. This allows to identify the relative contribution of these extremes across different regions and crops, which can guide more targeted drought adaptation and enable better decision-making.* *The empirical analysis of direct biophysically-induced damages during droughts is done at the district (regional) level for rainfed agriculture for eight major field crops in Germany from 2016-2022."* |
| 2 | My second comment relates to the focus on droughts or droughts and other extreme weather events in the current manuscript. Having read your work again, I have to admit that I still feel the manuscript predominantly focuses on droughts and less on other extremes. You now acknowledge this in the limitations (lines 500-505), where you highlight that these other extremes are not considered in the counterfactual. Besides that, the vast majority of the results focus on droughts. Section 3.4 should be about droughts and other extremes, but I find it still to be dominated by droughts, with little mention of other extremes. Can you describe this a bit better in the main text? Figure 6 is clear, so you could build on that. | Agreed and revised: We have added the clarifying text building on Figure 6 in section 3.4 and discussion section to better highlight the role of extreme events beyond drought in our manuscript. Pg 15, lines 447-451: *"Beyond drought and heat, Figure 6 also highlights the influence of other extreme events on crop yield anomalies in Germany. For example, black frost had notable effects on winter crops in 2021 and 2022 and alternating frost adversely affected rapeseed during these years. In contrast, waterlogging appears to have had a beneficial effect yield anomalies for most crops. These results show the complex interplay of weather extremes and their varying combinations, which determine the extent of yield losses from compounding and overlapping events in different years, as captured by the yield model."* Pg 17, lines 504-506: *"While drought and heat dominate the impacts, the yield model also captures the effects of other extremes—such as frost and waterlogging—whose contributions vary by crop and year."* Additionally, in response to Reviewer#2's comment, we have added a new bar plot in Figure 3, which also helps address this concern by illustrating the relative contribution of both drought and other hydro-meteorological extremes to total biophysical damages. This addition highlights the role of other extremes in our damage assessment. |

Minor comments

| S.No. | Reviewer's Comment | Authors' Response |
|---|---|---|
| 1 | You often refer to "regions" (e.g. line 187) and sometimes to "districts" (e.g. lines 227-229). Can you define somewhere what you mean by these? Do regions consist of districts (i.e. a region is bigger than a district), or are they synonyms? | Agreed and revised: Thank you for pointing out the inconsistent usage of "regions" and "districts." In our revised manuscript, we have aimed to maintain consistency by referring to them as "districts" and clarifying where necessary that these terms are used interchangeably. |
| 2 | Figures 3, 4, and 5: Check the legend and be consistent. Figures 3 and 5 refer to "drought losses," while Figure 4 refers to "drought damages." Should this be consistently "damage"?
- Figure 4: "mn" should be "millions." | Agreed and revised: We have revised the figure legends so that all references consistently use "damages," and replaced "mn" with "millions" in Figure 4 for clarity. |
| 3 | Lines 395-400: I would refrain from referring to specific districts without specifying where in Germany these districts are located. For non-German readers, that is hard to understand. Specify where Mecklenburg-Vorpommern, Lower Saxony, and Saxony-Anhalt are located. | Agreed and revised: Thank you for highlighting the need to situate these districts for non-German readers. In the revised manuscript, we have added brief geographical references so that readers unfamiliar with German state locations can better understand where these districts lie. Pg 14, lines 419-422: *"Interestingly, in 2019, 2020, and 2022, only limited losses were observed for sugar beets in Mecklenburg-Vorpommern (northeast Germany) and the bordering districts of Lower Saxony (northwest Germany) and Saxony-Anhalt (east-central Germany), despite a considerable share of area in these regions dedicated to growing this crop."* |
| 4 | Lines 408-410: "Contrary to intuition,… specific crop affected." Explain what extremes have a positive effect on yield anomalies. | Agreed and clarified: We have included specific examples (precipitation scarcity in March, heavy rain in July) in the revised manuscript to clarify how these extremes may sometimes benefit certain crops. Pg 14, lines 434-437: *"For example, precipitation scarcity in March was found to benefit spring barley, rapeseed, and winter barley if soils still hold sufficient winter moisture (Gömann et al., 2015). Similarly, heavy rainfall in July may increase yields for summer crops such as potatoes and silage maize, by mitigating drought stress in late summer when soils tend to be dryer (Samaniego et al., 2013)."* |

| S.No. | Reviewer's Comment | Authors' Response |
|---|---|---|
| 5 | Lines 430-435: Thanks for running all these robustness checks and sensitivity analyses. Can you add a couple of lines explaining these findings? What explains the lower and/or upper range? And how do these results increase your confidence in your main model specification? | Agreed and clarified: We have now added explanatory text in section *3.5 "Sensitivity analysis of estimated biophysically-induced direct damages"* clarifying how changes in the counterfactual period or the drought-area threshold affect the expected revenue benchmark, which in turn shifts the damage estimates. We also highlight that, despite these variations, our main results remain within the observed range of outcomes, demonstrating the robustness of our estimations. |
| 6 | Lines 506-508: "This study presents a conceptual framework … in agriculture." You just removed the conceptual framework when you revised the paper. Maybe rephrase it to "provides an empirical illustration"? I would also change "economic impacts" to "economic damage" in that sentence to maintain consistency throughout the paper. | Corrected and proofread: Thank you for noting the error. We have removed the mention of the conceptual framework from this sentence and have carefully proofread the text to ensure no such mentions remain. |
| 7 | You suggest future research both at the end of the discussion (lines 504-505) and at the end of the conclusions (lines 531-535). I suggest merging these. Either at the end of the discussion or the conclusion is fine. | Agreed and revised: We have merged the text on future research from the discussion with the relevant text in conclusion to maintain consistency. |

**Response to Reviewer#2**

| Reviewer's Comment | Authors' Response |
|---|---|
| I commend the authors for addressing all other major comments, but the issue of damage estimates including more than droughts still remains in my opinion. A simple figure illustrating estimated drought damages during NON DROUGHT years would resolve this. If damage in non drought years is 0, then what is present in the paper should not be changed. If however, the deviations are significant, then drought has either been over or underestimated. | Agreed and clarified: We appreciate the reviewer's continued attention to the accuracy of our damage assessment. We agree that some bias may arise due to the inclusion of multiple hydro-meteorological extremes in the yield model, even in non-drought years, which can slightly influence the estimation of expected revenues. However, we also note that excluding these extremes could underestimate total damages in drought-affected regions, as heat and precipitation scarcity often co-occur with drought and contribute to total damages as shown in the new bar plot introduced in the results section. We now discuss this trade-off in both the results and discussion sections. Specifically, we estimate the relative contribution of drought and other extremes to total biophysical damages (modified |

| Reviewer's Comment | Authors' Response |
|---|---|
| | figure 3) and assess the potential for bias by examining estimated damages in non-drought affected districts (Supplementary Figure S7). These additions strengthen the contribution of our approach while acknowledging its limitations.

Pg 12, lines 364-384: *"These biophysically-induced damages include the effects of all hydro-meteorological extremes, as captured by the LASSO yield models. Because the model accounts for multiple extremes, it is not possible to isolate the effects of drought alone from these damage estimates. To address this, we estimate the relative contribution of individual hydro-meteorological extreme to total damages using the average feature contributions (in percentage) to predicted yield anomalies, based on the models coefficients. These contributions are then used to weight total simulated damages, allowing us to approximate the share of revenue losses linked to individual extremes such as drought (Figure 3b). In years with high damages —2018, 2019 and 2022 —drought accounts for the largest share of total biophysical damages. Notably, heat and precipitation scarcity also contribute substantially during these years. This co-occurrence suggests that these extremes do not act in isolation and most likely interact with each other. For example, heat and precipitation scarcity may exacerbate the impacts of drought by adding further stress on crops or drought conditions may amplify the negative effects of high temperatures or low rainfall. This underscores the importance of including multiple hydro-meteorological extremes in the assessment of damages in drought-affected regions.*
*However, it is important to note that our approach also leads to positive values for biophysically-induced damages in districts and years that are not classified as drought (Supplementary Figure S7). This is because the yield model includes multiple extremes, which may still influence the non-drought years used to estimate expected yield and revenue in equation 2. While this may introduce small biases, they are not large in magnitude. Our approach may therefore slightly overestimate damages in* |

| Reviewer's Comment | Authors' Response |
|---|---|
|  | *droughts, whereas excluding the effects of other extremes would likely underestimate the total impacts in drought-affected districts and years. The true damages likely fall between these two cases. By including multiple hydro-meteorological extremes, our approach captures the biophysical effects related to extremes more comprehensively in drought affected regions. We further demonstrate the robustness of these estimates through sensitivity analyses that test alternative counterfactual periods and drought classification thresholds in Sect. 3.5."* |
|  | Pg 17, lines 488-492: *"Since the yield model includes all hydro-meteorological extremes, the non-drought years used to estimate expected revenue may still be influenced by these extremes. This can introduce small biases in the damage estimates. To address this, we use a five-year average of non-drought years, which helps smooth fluctuations and reduce the influence of other anomalies. The results are robust to alternative definitions of the counterfactual baseline which supports the reliability of our approach."* |

[Figure]

[Figure]

**Helmholtz Centre for Environmental Research – UFZ**
Permoserstr.15 · 04318 Leipzig · Germany

Dr. Giulia Vico
Editor
Natural Hazards and Earth System Sciences

**Mansi Nagpal**
PhD Student
Department of Economics
Fon +49 341 6025-4679
mansi.nagpal@ufz.de

Leipzig, 3/3/2025

**Cover letter for manuscript number EGUSPHERE-2024-2585**

Dear Dr. Vico,

Thank you for giving us the opportunity to submit a revised version of our manuscript **"Measuring extremes-driven direct biophysical impacts in agricultural drought damages"**. We appreciate the time and effort you and the reviewers invested in providing valuable feedback on our paper. The insightful comments and suggestions from you and the reviewers have been carefully integrated into the manuscript. Below is a brief overview of the main changes made to address the reviewers' and your comments:

(1) The introduction has been revised and enhanced to explicitly state the aim of the study and clarify the focus and scope of the investigation.

(2) The standalone section on conceptual framework has been removed to incorporate the recommendation to streamline the first part of the paper and maintain focus on empirical analysis of direct biophysically-induced damages.

(3) The structure of methodology section is enhanced with the introduction of a new subsection, *2.1 Overview of Analytical Approach* to provide a structured overview of the methodological components used in the study.

(4) The subsection 2.2 on damage measurement is improved with clarified definition of counterfactual, the role of statistical yield model and the use of drought-year prices to isolate and quantify the direct biophysically-induced damages.

(5) In the results section, we have added a new section 3.5 on robustness checks as sensitivity analysis, to show the extent of damage driven by expected revenues as counterfactuals.

**Helmholtz Centre for Environmental Research – UFZ**

Company domicile: Leipzig

Permoserstr. 15, 04318 Leipzig, Germany
or
PF 500136, 04301 Leipzig, Germany

info@ufz.de
www.ufz.de

Registration court: Leipzig district court
Commercial register No. B 4703

Chairman of the Supervisory Board:
MinDirig'in Oda Keppler

Scientific Director:
Prof. Dr. Rolf Altenburger

Administrative Director:
Dr. Sabine König

Bank details:
HypoVereinsbank Leipzig
Sort code 860 200 86
Account No. 5080 186 136
Swift (BIC) code HYVEDEMM495
IBAN No. DE12860200865080186136
VAT No. DE 141 507 065
Tax No. 232/124/00416

(6) The discussion section has been significantly improved to elaborate on methods and their limitations, compare findings with studies beyond Germany and discuss broader implications of our results.

(7) Further edits have been made throughout the manuscript to incorporate your feedback and address the specific comments provided by the reviewers.

Please find detailed response to the specific comments by you and the reviewers in the document below. We believe that our revisions have addressed your and reviewers' concerns and suggestions, resulting in a much improved paper.

Thank you for considering our revised manuscript. We look forward to your response.

Best regards,
Mansi Nagpal on behalf of the authors

Manuscript Number: **egusphere-2024-2585**

**Response to Editor and Reviewers comments**

3 March 2025

**Contents**

**Editor and reviewer comments and authors' replies marked in black**

**Previous manuscript and supplement text marked in brown**

**New manuscript and supplement edits marked in blue**

All page and line numbers refer to the revised, unmarked manuscript

**Response to Editor**

| S.No. | Editor's Comment | Authors' Response |
|-------|------------------|-------------------|
| G1.a | The reviewer have provided very insightful comments.
I agree with the reviewers that, in its current form, the manuscript does not clarify upfront what is being investigated. The first part is particularly confusing and it is difficult for me to evaluate whether the planned changes will sufficiently improve the clarity. | Agreed and clarified: We acknowledge the need to clarify the study's focus from the outset. To address this, we have revised and extended the text around line 71 in the introduction to explicitly state the aim of this study and introduce a clear definition of direct biophysically-induced damages to clarify the focus of our analysis.
Pg 2, lines 71-77: *"In this study, we address this bias by assessing the economic damage of drought in combination with concurrent or successive weather extremes in rainfed agriculture. The aim of this study is to measure the direct biophysical damage of extreme hydro-meteorological drivers during droughts (hereafter called direct biophysically-induced damages) and assess their contribution to farm revenue losses. These damages refer to the loss in revenue caused by the effects of extreme hydro-meteorological drivers on crop yields, without accounting for other economic impacts, such as changes in costs. They include the effects of droughts themselves, as well as additional damage from concurrent or successive weather extremes that exacerbate drought-related effects in regions experiencing drought conditions."*

To further streamline the manuscript and improve its clarity, we have:
  (a) Removed the standalone conceptual framework section: The original conceptual framework illustrated the full range of impacts, including both direct damages and indirect economic impacts. To maintain focus on empirical investigation of direct biophysically-induced damages, we have removed references to the old conceptual framework in the introduction and eliminated the standalone Section 2 on conceptual framework. Please refer to our detailed response to your comment |

| S.No. | Editor's Comment | Authors' Response |
|---|---|---|
| | | G1.b for further details. |
| | | (b) Introduced new subsection: To enhance the structure of methodology section, we have introduced new subsection *"2.1 Overview of Analytical Approach"* that outlines the methodological components used in the study and guide the reader's understanding of our approach. Please refer to our detailed response to your comment G1.b for further details. |
| | | (c) Standardized terminology throughout the manuscript: In response to Reviewer 1, comment G2, we now consistently use "direct biophysically-induced damages" instead of "economic impacts" to more accurately reflect the focus of our study. |
| G1.b | I add to the reviewers comments that the role of the contextualization framework (section 2 and Fig 1) in the general economy of the manuscript remained unclear to me. Either it is better integrated in the work or, likely better, it is removed from the manuscript, thus focusing the work to the actual analyses. More in general, I urge the authors to critically check the logic behind presenting the material in a certain order and consider if and where clarifications are needed. | Agreed and revised: Thank you for highlighting an important area for improving the coherence of the manuscript. In response, we have removed the standalone section on conceptual framework from the revised manuscript along with figure 1. This has allowed us to restructure the manuscript along a more standard format, with section 2 now titled "*Methodology*".

To retain key theoretical components from the removed section, we have introduced a new subsection, titled *"2.1 Overview of Analytical Approach"* [See pg 3], within the methodology. This new section provides direct link between the theoretical background and the methodology used in the study. It outlines the key methodological components used to estimate direct biophysically-induced damages and introduces the causal pathways by which hydro-meteorological extremes impact crop yields and result in revenue losses. These explanations are structured to support the understanding of the empirical methods, guiding the reader to relevant methodological components in the subsequent sections and improving the |

| S.No. | Editor's Comment | Authors' Response |
|---|---|---|
| | | logical flow of the manuscript. As a visual aid and to enhance clarity, we have replaced the old figure 1 with a new and more concise figure to present these connections between casual pathways with the analytical approach used in the paper. These changes help to clarify the structure of the information presented in methods section and guide the reader's understanding of our approach. Additionally, we have ensured that the explanations within this new section are directly tied to the scope and objectives of the study by reducing the theoretical discussion on indirect impacts of droughts, which are not assessed in this paper and emphasizing on direct biophysically-induced damages, which form the core of our empirical analysis. |
| G1.c | Furthermore, the authors should clarify why several extremes are defined (Table 1) but the focus appears to be on droughts only. | Agreed and clarified: We have clarified in the methodological description of section 2.3 the use of several extremes defined in table 1 and its use in estimating economic damage driven by the biophysical impacts of droughts and their interaction with other extremes. Pg 8, lines 262-267: *"To simulate crop yields (in decitons per hectare - dt/ha), we multiply the predicted yield anomaly by the district-level mean yield. This approach allows us to isolate crop yields attributable to hydro-meteorological extremes defined in Table 1, including droughts. These simulated yields are then used for damage assessment in drought-affected regions categorised using the SMI (as described in next section), aligning with the objective of quantifying the economic damages during droughts driven by the biophysical impacts of droughts and their interaction with other extremes."* |
| G1.d | Some key methodological issues have also been raised and the authors have delineated apparently satisfactory ways to address some of the main concerns. Of particular relevance are 1) clarifying the definition of counterfactual and showing that the results are robust to any somewhat subjective choice in its definition; and 2) acknowledge | We greatly appreciate the valuable feedback of the reviewers and acknowledge the recognition of our efforts in addressing their concerns. |

| S.No. | Editor's Comment | Authors' Response |
|---|---|---|
| | the limitations of the methods and discuss their implications, in particular regarding the role of detrimental conditions beyond water stress. | |
| G1.e | Beyond the points raised by the reviewers, I notice that the period 1999-2022 is a rather short one to identify extremes. Moreover, in principle all definition of extremes are not justified and the threshold tested. | Agreed and clarified: We acknowledge the concern regarding the length of the period. To address this, we have included temporal histograms for all extremes for the maize crop as a representative crop in the supplementary material, which demonstrates that the 1999–2022 period captures a substantial number of extreme events, notably the exceptional droughts of 2003, 2018-2020, and 2022, waterlogging in 2001, 2007, 2010 and 2013, as well as severe frost and heat events.
 Pg 8, lines 258-262: *"To illustrate the adequacy of the 1999–2022 period in identifying extremes, temporal histograms of all extreme weather events for the maize crop, used as a representative crop, are provided in the supplementary material (**Supplementary Figures S2-S3**). These histograms demonstrate that the selected period captures a substantial number of extreme events, notably the exceptional droughts of 2003, 2018-2020, and 2022, waterlogging in 2001, 2007, 2010 and 2013, as well as severe frost and heat events."*

 Regarding the justification and testing of thresholds, we have clarified in the discussion that, as we relied on an established statistical model with pre-defined thresholds, their sensitivity was not assessed. This has been acknowledged as a limitation of the study.
 Pg 17, lines 494-498: *"Second, the estimation of revenue losses might be underestimated due to the inherent limitations of the statistical yield model in simulating extreme crop yields. This underestimation partially arises from the use of pre-defined thresholds for extreme events. Since the study relied on an established statistical model, we did not assess the sensitivity of these thresholds, which should be explored in future research to improve robustness."* |

| S.No. | Editor's Comment | Authors' Response |
|-------|------------------|-------------------|
| G1.f | I further add that the discussion needs substantial development, to include discussion on the methods and their limitations, comparisons of results beyond Germany only, and the implications of the main conclusions. | Agreed and enhanced: We have significantly improved the discussion section by addressing the methods and their limitations, comparing our results with studies from regions beyond Germany and elaborating on the implications of our main conclusions. Additionally, we have expanded the abstract and the conclusion to include an outlook on integrating damage estimates into drought monitoring systems to enhance early warning and adaption.

 Pg 17, lines 492-502: *"While our estimates provide robust insights into the biophysical damages of droughts and associated extremes in drought affected regions, there are some limitations to consider. First, our analysis is focused on short-term impacts damages and does not include adaptation costs or indirect impacts beyond the immediate consequences of biophysically induced yield losses. Second, the estimation of revenue losses might be underestimated due to the inherent limitations of the statistical yield model in simulating extreme crop yields. This underestimation partially arises from the use of pre-defined thresholds for extreme events. Since the study relied on an established statistical model, we did not assess the sensitivity of these thresholds, which should be explored in future research to improve robustness. Last, this yield model is based on anomalies relative to district-level means which limits our ability to fully control for the biophysical impacts of weather extremes in the counterfactual. While a non-extreme weather events counterfactual could have provided valuable insights into the interplay between droughts and other extremes, this was not feasible within the current modelling framework. Future research should focus on testing different types of yield models that allows control of impacts of weather extremes in the counterfactual while capturing the dynamics of extreme weather impacts on yields."*

 Pg 17, lines 481-490: *"In comparison to our* |

| S.No. | Editor's Comment | Authors' Response |
|---|---|---|
| | | *findings, García-León et al. (2021) estimated that agricultural losses due to droughts in Italy ranged from €0.55 billion and €1.75 billion per year, while Howitt et al. (2015) reported crop revenue losses in California, United States of approximately $902 million to $940 million per year. Our result that maize was the most effected crop during recent droughts in Germany is consistent with the findings of Brás et al (2021), who found maize as experiencing the highest production losses among cereals across Europe due to droughts and heatwaves between 1964 and 2015. Maize's vulnerability to drought is not limited to Europe. In the United States, substantial yield variability in maize has been linked to drought and heat stress (Zipper et al., 2016). Similarly, in China, maize yield losses have been shown to increase with the severity of drought, contributing to significant reductions in maize production across the country (S. Liu et al., 2022). These comparisons highlight the dual challenge of mitigating economic losses across diverse cropping systems and addressing the specific vulnerabilities of drought-sensitive crops like maize. They underscore the importance of globally coordinated efforts to enhance agricultural resilience in the face of increasing weather extremes."*

Pg 16, lines 447-448: *"The spatially distributed approach used here can be adapted in other regions to provide more precise assessment of revenue losses and to inform policy planning."*

Pg 16, lines 458-459 *"These findings underscore the need for spatially targeted polices and interventions, particularly in northern and eastern Germany, where agriculture is disproportionally affected during droughts."*

Pg 17, lines 466-467: *"It helps disentangling the contributions of extreme hydro-meteorological drivers of yields vis-à-vis other* |

| S.No. | Editor's Comment | Authors' Response |
|-------|------------------|-------------------|
| | | *drivers of yields to revenue losses, underlining the importance of these factors in shaping agricultural outcomes."*

 Pg 1, lines 28-30: *"Future integration of routine drought damage estimation into operational monitoring and forecasting systems would enhance early warning capabilities, improve economic preparedness against increasing weather extremes, and support more proactive adaptation strategies."*

 Pg 18, lines 528-533: *"Future work should focus on routinely estimating these losses within operational drought monitoring systems such as the German Drought Monitor (Zink et al., 2016), and forecasting frameworks like Hydroclimatic Subseasonal-to-Seasonal forecasting system (Hydroclimatic Forecasting System, 2024). By linking hydro-meteorological variables with projected economic damages, such integration would enhance early warning capabilities, improve economic preparedness against increasing weather extremes, and support more proactive adaptation strategies."* |

**Additional minor comments:**

| S.No. | Editor's Comment | Authors' Response |
|-------|------------------|-------------------|
| 1. | L 48: I'd say drought types or similar, not classifications | Agreed and revised: We have replaced "classifications" with "types" in the text to better align with the intended meaning. |
| 2. | L 61: weather extremes can mean many things: I suggest adding a few examples | Agreed and clarified: We have added the following examples to clarify the meaning of weather extremes.
 Pg 2, lines 61-64: *"For example, extreme heat during summer droughts can intensify damage to crops such as maize, further reducing yields (AghaKouchak et al., 2014). Similarly, winter crops like wheat can suffer significant losses from drought, heat and drought followed by periods of excessive rainfall, negatively affecting yields and* |

| S.No. | Editor's Comment | Authors' Response |
|---|---|---|
| | | *harvest quality (J. Ding et al., 2018; Zampieri et al., 2017)."* |
| 3. | L 66-71: the difference between the two types of damage/impacts is rather obscure | Agreed and clarified: Thank you for your comment. The term "impact" has been removed from these lines, as it was originally used in reference to the conceptual framework, which has now been removed during the revision process. We now use only the term "damage," which has been elaborated on in the revised text to ensure clarity.
Pg 2, lines 72-75: *"The aim of this study is to measure the direct biophysical damage of extreme hydro-meteorological drivers during droughts (hereafter called direct biophysically-induced damages) and assess their contribution to farm revenue losses. These damages refer to the loss in revenue caused by the effects of extreme hydro-meteorological drivers on crop yields, without accounting for other economic impacts, such as changes in costs."* |
| 4. | L 223: "it" instead of them | Agreed and revised: Thank you for pointing this out; we have replaced "them" with "it" as suggested. |

**Response to Reviewer#1**

| Reviewer's Comment | Authors' Response |
|---|---|
| This paper explores the economic impacts of multiple climate extremes, focusing on droughts, by estimating revenue changes. The economic damage is defined as the difference between expected and actual revenues. Using a counterfactual that compares expected revenues to realised revenues under drought conditions, the economic impact of droughts is estimated. The topic is timely and relevant to the journal, but I would like to offer a few suggestions that I believe are important to take on board.
One potential concern is the definition of economic impact as the difference between realized and expected revenues. This approach means that a significant portion of the estimated economic impact depends on how expected revenues are defined. You base the **counterfactual** (expected revenues) on past non- | Thank you for appreciating the relevance of our contribution and providing valuable comments on how to improve this manuscript.
We have carefully reviewed these comments and have made significant revisions to address them, summarized below:
• Counterfactual & robustness checks: We have clarified the definition and included robustness checks to show the extent of damage driven by expected revenues as counterfactuals. See our response to *general comment G1* for further details.
• Definition of economic damages: We clarified that our focus is on assessing direct biophysically-induced damages of extreme hydro-meteorological drivers during droughts and their contribution to farm revenue losses. |

| drought revenues within the same region. I am uncertain if this is the best approach, and we might have taken different directions here. To address this, a clear justification for your counterfactual is needed, likely supported by **robustness checks** to show how results might change with different counterfactuals. Additionally, I would like to discuss (i) your **definition of economic impacts** and (ii) **whether droughts and climate extremes are best measured dichotomously or continuously**. These three points form the basis of my general comments. I have also provided a **few minor suggestions** and textual edits below. | For specific details, please refer below to our response to *general comment G2.*
• Continuous measurement of droughts and extremes: We clarified that the extremes including droughts are measured as continuous variables in statistical yield model. However, drought occurrences are categorized dichotomously (including spatial and temporal development characteristics) to focus damage assessments on affected regions and for counterfactual estimations. Please refer to our response to *general comment G3* for details.
• Aim of study: The study's aim is now explicitly stated in the revised introduction. Please see our response to *specific comment S1* for the details.
• Farm level damage assessment literature: We have added literature on farm-level damage assessment for a more comprehensive introduction. Please see our detailed response to *specific comment S2* for further details.
• Conceptual figure: The color scheme of the conceptual figure has been revised with distinct grayscale tones to ensure that the biophysical and economic processes remain distinguishable when the document is printed in black-&-white. In the revised manuscript, this figure has been moved to the supplementary materials (now Supplementary Figure S1). The new figure 1, also designed with distinct grey scale tones, presents a concise overview of the empirical analysis and the methods used. |
|---|---|

**General comments**

| S.No. | Reviewer's Comment | Authors' Response |
|---|---|---|
| G1.a | My main suggestion is to reconsider your counterfactual and clarify what it is actually measuring. How do you accurately estimate the expected revenues? What is the counterfactual representing? Defining a reliable counterfactual is critical because the economic impacts in your paper are defined as the difference between observed and expected revenues. Currently, you define expected revenues as the average revenues over the past five non-drought years. However, I am uncertain about whether this counterfactual is | Agreed and enhanced: To clarify the aim of the counterfactual and what it represents, we have modified and added the following text:
Pg6, lines 185-195: "*The counterfactual conditions aim to represent the average non-drought conditions specific to each region. In the context of ongoing climate variability, it is critical that the counterfactual conditions represent the evolving regional climatology (Suarez-Gutierrez et al., 2023) rather than relying on an idealized "normal" year in the traditional sense, which may no longer occur in practice. In this analysis, we define the counterfactual* |

| S.No. | Reviewer's Comment | Authors' Response |
|---|---|---|
| | consistently measuring the same expectations across regions, especially since no other observable factors are considered. As noted in lines 156-173, the counterfactual seems somewhat arbitrarily defined.

 For example, consider two regions where neither has experienced a "normal" year during the reference period. Region 1 has had consecutive slightly wet years, while region 2 has had five consecutive slightly dry years (though not extreme). Consequently, your expected revenues for region 1 are based on slightly wet conditions, while for region 2, they reflect slightly dry conditions. As a result, the estimated economic impact of droughts is now being benchmarked against two different baselines, which could affect the accuracy of your estimates. | *conditions as the average conditions in the preceding five non-drought years. We selected a five-year window following Trenczek et al. (2022), who used it to estimate damages for 2018 and 2019 droughts in Germany. The reason for this number of years is a trade-off: using more years could in theory further enhance the statistical representativeness regarding local climatic conditions, but it risks introducing bias by masking changing market and production conditions, as well as the overall trend in climate change, which also influence local yields and revenues (Lobell et al., 2011).*
 *We determine drought (and non-drought) years based on the soil moisture. In order to do so, we use the Soil Moisture Index (SMI) metric, as explained in Sect. 2.4, and exclude any drought years in the average estimation, an improvement over existing approaches in the literature."*

 Additionally, we have clarified our presentation for readers to address the reviewer's related concern about whether the observable factors considered in the counterfactual design are sufficient to reflect spatial differences between regions. To this end, we have modified and added the following text:
 Pg9, lines 279-281: *"Using monthly SMI data, at a resolution of 4km x 4km and covering the Germany entirely, the monthly average area under drought conditions was estimated (Nagpal et al., 2024) for each district. The drought categorization based on the SMI reflects regional differences in climatic conditions as the SMI is calculated relative to the local historical soil moisture distribution in each district."* |
| G1.b | Then, a second objective of the paper is to investigate the economic impacts of the interplay of droughts and extreme weather events. I do not yet see how this is reflected in your current counterfactual, as those extreme weather events are not considered when you | Agreed and enhanced: We have added the suggested robustness checks as sensitivity analysis, now detailed in a new results section *3.5 "Sensitivity analysis of estimated* |

| S.No. | Reviewer's Comment | Authors' Response |
|---|---|---|
| | define your counterfactual. The implications of this are that the expected revenues do not consider any past exposure to other extreme weather events, making me wonder how accurate your economic impact estimates are. One way forward to convince me that your counterfactual is measuring what it intends to measure is to include robustness checks, with different counterfactual definitions (e.g., using shorter or longer reference periods, or incorporating multiple extreme weather events). Alternatively, you could consider defining your counterfactual based on matching or regression-based approaches, which allows you to account for observable characteristics such as the severity of drought (using continuous measures like soil moisture index), crop types, or land area. It would also be useful to indicate how much of the estimated economic impact is driven by the occurrence of droughts versus changes in the expected revenues themselves (i.e. how do your results change when defining different counterfactuals?) | *biophysically-induced direct damages"* [See page 15], of the revised manuscript. The new sensitivity analysis include the following:

(a) Varying the counterfactual period by $\pm$ 1 year to examine the effect of different reference periods on the estimates.
(b) Adjusting the drought classification criteria by testing thresholds with $\pm$ 5% variations in the area of each district with an SMI < 0.2 per month, in addition to the original 20% threshold.

We have also clarified in the manuscript text how our counterfactual address potential bias from exposure to other extreme events, as well as the limitation of our approach.
Pg7, lines 197-200: *"While the counterfactual is designed to exclude drought years, it is possible that some exposure to other extremes could still be reflected in the yields of non-drought years. Any potential yield anomalies in non-drought years, which could lead to over- or under-estimating drought damages, are addressed through the approach of estimating expected revenue based on the five-year average. This helps to smooth out any random yield fluctuations and minimize the influence of non-drought related anomalies."*
Pg 17, lines 498-501: *"Last, this yield model is based on anomalies relative to district-level means which limits our ability to fully control for the biophysical impacts of weather extremes in the counterfactual. While a non-extreme weather events counterfactual could have provided valuable insights into the interplay between droughts and other extreme weather events, this was not feasible within the current modelling framework."* |
| G2. | Are you truly estimating the economic impact of droughts? Your analysis focuses on changes in revenues, but it does not account for | Agreed and clarified: We agree that our analysis focuses on revenue changes rather than a full economic impact assessment, which |

| S.No. | Reviewer's Comment | Authors' Response |
|-------|-------------------|-------------------|
| | changes in costs (e.g. inputs, intermediates etc.). I could live with damage but feel like you are not estimating economic impacts. | would require accounting for costs such as inputs and operations. To address this, we have clarified in the introduction that our focus is on assessing direct biophysically-induced damages of extreme hydro-meteorological drivers during droughts and their contribution to farm revenue losses.

Pg2, lines 71-74: *"In this study, we address this bias by assessing the economic damage of drought in combination with concurrent or successive weather extremes in rainfed agriculture. The aim of this study is to measure the direct biophysical damage of extreme hydro-meteorological drivers during droughts (hereafter called direct biophysically-induced damages) and assess their contribution to farm revenue losses."*

Additionally, we have revised the manuscript to consistently communicate that we are estimating direct biophysically-induced damages, rather than attempting a full economic impact assessment.

a. We have modified the old conceptual framework figure (now part of a new section *2.1 "Overview of Analytical Approach"*) to emphasize the specific component of direct biophysically-induced damages as the focus of our measurement and analysis. Please refer to improved figure1 (page 4) in the revised manuscript.
b. We have now consistently used the term "direct biophysically-induced damages" instead of "economic impacts" to more accurately reflect the scope of our analysis.
c. We have ensured that there is always a qualifier clarifying the meaning of the word "impacts" and prevent any misunderstanding as referring to economic impacts.
d. We have revised the figure legends in the results section to clarify that they pertain to damages. |

| S.No. | Reviewer's Comment | Authors' Response |
|---|---|---|
| G3. | Are droughts something to be measured dichotomously? Same for the extreme weather events. There seems to be a slight mismatch between the research gap you identify and your approach in practice. For example, in lines 43-45, you describe the research gap as focusing on the variability and intensity of droughts. This suggests a continuous definition, where drought ranges from slightly dry to extremely dry conditions. However, if I understand correctly, in your paper droughts are defined dichotomously—either present or absent. The same issue arises in lines 58-59. Is the research gap you have identified (regarding the variability of droughts and extreme weather) truly being addressed by your current approach? | Agreed and clarified: To address the reviewer's concern regarding how our dichotomous drought categorization methodology, essential for defining the counterfactual, accounts for the complexity of drought occurrence including variability and intensity of droughts as described in lines 43–45, we have revised section 2.4 to include the following clarification:

 Pg9, lines 284-290: *"This approach accounts for the slow development and spatial and temporal accumulation characteristics of droughts. By using a threshold of SMI<0.2, we comprehensively capture all regions affected by droughts, including those experiencing varying intensities from severe (SMI<0.1) to exceptional conditions (SMI<0.02). This method enables the identification of non-drought years necessary for estimating expected revenues under counterfactual conditions. To evaluate the effect of this drought classification approach on damage estimates, we conducted sensitivity analyses by varying the threshold for the proportion of affected area (±5%), to confirm the robustness of the damage estimates under alternative drought classification criteria."*

 To address the reviewer's concern regarding the variability and intensity of droughts as described in lines 58-59, it is important to clarify that while thresholds are used for categorizing regions as drought (non-drought), the statistical yield model incorporates it, along with other extreme weather events, as continuous variables. This approach accounts for their severity, where higher intensities leads to greater predicted yield reductions. We have revised the manuscript to reflect this clarification.
 Pg 8, lines 247-253: *"These indicators are calculated by counting the days in a month that exceed or fall below the defined thresholds.[.......] All features are used as* |

| S.No. | Reviewer's Comment | Authors' Response |
|-------|--------------------|--------------------|
|       |                    | *continuous variables to account for stronger effects on crop yields through more intense extremes."* |

**Specific suggestions**

| S.No. | Reviewer's Comment | Authors' Response |
|-------|--------------------|--------------------|
| S1. | I have read your introduction but couldn't identify the aim of the paper. I could be wrong here but my feeling is that lines 66-70 intend to do this. It is a little vague and would help me if you make this more concrete. I am looking for a sentence like "The aim of this paper is to…." or "This paper addresses the question…." | Agreed and clarified: We have revised the text surrounding lines 66-67 (now lines 71-72) to explicitly clarify the aim of our study for the benefit of the readers.
 Pg2, lines 73-74: *"The aim of this study is to measure the direct biophysical damage of extreme hydro-meteorological drivers during droughts (hereafter called direct biophysically-induced damages) and assess their contribution to farm revenue losses."* |
| S2. | Lines 83-85: Perhaps you could consider adding some studies on farm-level economic damage to be complete. There is a lot of ongoing work here on adaptation literature but also on estimating drought damage on the farm level. | Agreed and enhanced: As suggested, we have incorporated the following text in the introduction of the paper, which discusses the added references on recommended empirical studies analyzing drought damages at the farm level.
 Pg3, lines 88-94: *"Alternatively, there are several empirical studies analysing drought damages at the farm level that often incorporate adaptation strategies (van Duinen et al., 2015; Wens et al., 2021), input changes (Prasanna, 2018) and factors affecting localized responses to droughts (Ahmad et al., 2022; Garbero & Muttarak, 2013; Gray et al., 2009). Their empirical findings are tailored to specific context and may not be readily scalable to broader regions. Conversely, national-level assessments, though comprehensive, fail to capture the spatial variability of drought impacts. As droughts can vary greatly across different locations and times (Jaeger et al., 2013; Samaniego et al., 2013), there is a need for consistent, spatially-explicit damage assessments (Meyer et al., 2013) bridging the gap between farm-level-detail and national-level scope."* |

| S3. | Figure 1: I printed your manuscript in black and white and could not see any colour differences. Consider changing the colours or thinking of some other way to underline what is a biophysical process and what is an economic process. | Agreed and modified: Thank you for the suggestion. We have replaced the original color scheme of figure 1 with distinct grayscale tones to ensure that the biophysical and economic processes remain distinguishable when the document is printed in black-and-white. Please refer to improved figure1 (page 4) in the revised manuscript. |
|------|------|------|

**Response to Reviewer#2**

| Reviewer's Comment | Authors' Response |
|---|---|
| The authors address the complicated question of accurately estimating the direct impacts of droughts on agricultural yields. In doing so, they tackle a number of issues that confound the drought estimates, including the co-occurrence of other extreme weather events, the regional heterogeneity in occurrences and effects that limit the viability of national aggregated measures and the presence of indirect effects that come from secondary and tertiary impacts. Using Germany as the backdrop, they find that the direct impact of droughts amounts to 781 million euros in the period investigated, accounting for 60% of reported yield losses in drought years, going as far as 97% of total damage when the focus is on rice yields in 2018. They also find a discrepancy when comparing national aggregated estimates to regionally estimated losses, suggesting a preference for regional estimates.

Some issues remain and are addressed below | Thank you for your appreciation of the significance of our contribution. We found your feedback valuable in further improving our manuscript and have made key revisions to our manuscript to address your comments, as outlined below:

• Focus of investigation: We explicitly stated the aim of this study in the introduction. For details, please refer to our response to *comment 1.*

• Damage measurement: We clarify the role of statistical yield model and assumption of constant prices in assessing direct biophysically-induced damages of hydro-meteorological extremes during drought years in the manuscript. We also present additional sensitivity analyses to evaluate the potential of over- or underestimation of drought damages. Please refer to our response to *comment 2* for the details.

• Use of current prices in damage assessment: We clarify that the inclusion of prices is essential to our aim of quantifying the direct biophysically-induced damage in monetary terms. The use of current prices reflects conditions contemporaneous to the drought and maintain consistency with previous studies. Please refer to our response to *comment 3* for more information.

• Simulation of yields using regression coefficients: We clarify the use of extreme events features of the LASSO model for simulating yields used in the damage assessment. Details are provided in our response to *comment 4*.

• Spatial disaggregation: We clarify crop-specific assessments as a consistent component of both national-level and regional-level analysis and the discrepancies observed in national estimates arising from spatial-disaggregation. Please see our response to c*omment 5* for further details.

• Typos corrected and proofread: We have corrected the typos and thoroughly proofread the text to ensure no additional errors remain. |

| S.No. | Reviewer's Comment | Authors' Response |
|---|---|---|
| 1. | The first issue I came across while reading was confusion on what exactly was being investigated. For the first few pages, I assumed the purpose was an investigation of the impact of agricultural droughts measured by soil moisture, but after a few pages, the phrase "extreme weather on agriculture during drought years" gave the impression that the investigation was a secondary effect of other extreme weather events during drought years. After reading, I am convinced that the paper is just about the impact of drought (first, with a combination of other extremes investigated in section 4.4), if I am mistaken, it adds to the confusion I had while reading through. Simplifying the text and stating precisely what was investigated would be ideal. | Agreed and clarified: To address the confusion regarding the focus of our investigation, we have explicitly stated the aim of the study and clarified it further to eliminate ambiguity, as detailed below. Pg 2, lines 71-79: *"In this study, we address this bias by assessing the economic damage of drought in combination with concurrent or successive weather extremes in rainfed agriculture. The aim of this study is to measure the direct biophysical damage of extreme hydro-meteorological drivers during droughts (hereafter called direct biophysically-induced damages) and assess their contribution to farm revenue losses. These damages refer to the loss in revenue caused by the effects of extreme hydro-meteorological drivers on crop yields, without accounting for other economic impacts, such as changes in costs. They include the effects of droughts themselves, as well as additional damage from concurrent or successive weather extremes that exacerbate drought-related effects in regions experiencing drought conditions. To isolate the biophysical impacts of these extremes on crop yields from other influencing factors, we employ crop specific statistical yield models. By comparing the direct biophysically-induced damages estimated from these models with reported farm revenue losses, we can identify the relative contribution of these factors across different regions and crops, which can guide more targeted drought adaptation and enable better decision-making."* We have also modified figure1 (pg 4) to emphasize the specific component of direct biophysically-induced damages as the focus of our analysis. |
| 2. | The measure of damage in equation 1 itself may be over or underestimating drought effects in its current form. With the impact being the difference between the expected revenue and the actual revenue, it ascribes this difference in its entirety to drought effects, which may not be entirely true. It is the classic diff-in-diff argument. For the damage equation | Agreed and clarified: Thank you for the insightful comment. In the revised manuscript, we have clarified the role of statistical crop yields and constant price assumption in ascribing the difference between expected and actual revenue to biophysically-induced impacts of extreme weather events including droughts. We have also clarified our approach |

| S.No. | Reviewer's Comment | Authors' Response |
|---|---|---|
| | to be solely due to droughts, the authors current approach would necessitate that in non-drought years, expected outcomes ALWAYS match the realized outcomes. I am doubtful that this is true, and as such, any shortfalls in non-drought years would imply that negative drought effects are overestimated while any windfalls (realized yields greater than expected) would underestimate the drought effects. Therefore, I suggest that the damage be estimated as $$D_t = \sum_{c=1}^{8} (\bar{R}_{expected,c,t} - R_{actual,c,t}) - \frac{1}{T} \sum_{t=1}^{T} \sum_{c=1}^{8} (\bar{R}_{expected,c,t} - R_{expected,c,t}^{ND})$$ Where the additional term is the average difference between expected revenue and realized revenue in T non drought years in the study. This way, any non-drought related discrepancies can be correctly accounted for. | to handling potential yield shortfalls or windfalls through the five-year window for estimating expected revenues in Equation 1, as detailed in the text below. Furthermore, we have tested the sensitivity of our approach by varying the counterfactual period by ± 1 year to assess the risk of over- or underestimating drought effects. The results are detailed in a new subsection *3.5 "Sensitivity analysis of estimated biophysically-induced direct damages"* [See pg 15]. Pg 7, lines 216-221: *"We use simulated crop yields to estimate actual revenue for drought years and expected revenue under counterfactual conditions for non-drought years, in order to calculate damages in eq.1. This ensures that the damage estimates are explicitly based on yield variability driven by EWE as described in equation 3, while excluding other factors unrelated to extreme hydro-meteorological drivers. Along with the assumption of constant prices, this methodology ensures that the revenue deviation between expected and actual revenues is attributed solely to the direct biophysically-induced yield impacts during droughts.* Pg 6, lines 197-200: *"While the counterfactual is designed to exclude drought years, it is possible that some exposure to other extremes could still be reflected in the yields of non-drought years. Any potential yield anomalies in non-drought years, which could lead to over- or under-estimating drought damages, are addressed through the approach of estimating expected revenue based on the five-year average. The helps smooth out any random yield fluctuations and minimize the influence of non-drought related anomalies."* |
| 3. | In equation 2, using the current price to estimate expected revenue might be problematic given that others have found that extreme weather events have their own distinct impact on prices (Berhanu & Wolde, 2019; Felix & Romuald, 2012; Ray, 202 1). It may be beneficial to use in year prices adjusted | Agreed and clarified: To clarify the use of drought-year prices for estimating expected revenues, we have provided the following explanation in the revised manuscript. Pg 6, lines 205-212: *"The use of drought-year prices to estimate expected revenues reflects contemporaneous market conditions during* |

| S.No. | Reviewer's Comment | Authors' Response |
|-------|-------------------|-------------------|
| | for inflation to estimate expected revenues. If the idea was to allow for the focus to be just on yields, then I would recommend just leaving prices out entirely. Including prices would mean that expectations are driven by two sources: expected yields and expected prices, both of which can be separately impacted by domestic and external weather shocks. | *the drought year and maintains consistency with previous studies. While using in-year prices for estimating expected revenues might capture the indirect effects of droughts on prices (Badolo & Somlanare, 2012; Berhanu & Wolde, 2019; C. A. Ray, 2021), it would also incorporate other agricultural market developments unrelated to local droughts or extremes, complicating the attribution of damages to regional extreme hydro-meteorological drivers. Holding prices constant ensures that the damage estimates focus solely on the yield changes induced by extreme hydro-meteorological drivers, providing an economic estimation of biophysically-induced direct damages in monetary terms."* |
| 4. | The statistical crop yield model shows a regression that included several weather extremes on the right-hand side, but did not discuss how the drought contribution to yield was extracted or what it in fact looks like. Some descriptive statistics would be helpful here. Is drought driven yield just beta*drought? Is the dependent variable in subsequent analysis yields as a result of droughts? More exposition on what exactly was done to generate the variable of interest would be ideal. | Agreed and clarified: As suggested, we have modified the methodological description of statistical crop yield model in section 2.3 and have added descriptive statistics in a new table-Appendix A (Page 18,19), to make it easier for readers to understand the model outputs without consulting the original publication (Heilemann et al., 2024). We have clarified that the dependent variable is indeed the yield anomaly as a result of droughts (and other extreme events).
 Pg 8, lines 256-268: "*Based on the extreme event features, the LASSO models predict the annual yield anomaly (in %) as the dependent variable, representing the deviation of yields from the district-level mean yield for 1999-2022. Details on the standardized coefficients of the crop-specific LASSO models can be found in Table S2 of Heilemann et al. (2024). To illustrate the adequacy of the 1999–2022 period in identifying extremes, temporal histograms of all extreme weather events for the maize crop, used as a representative crop, are provided in the supplementary material (Supplementary Figures S2-S3). These histograms demonstrate that the selected period captures a substantial number of extreme events, notably the exceptional droughts of 2003, 2018-2020, and 2022, waterlogging in 2001, 2007, 2010 and 2013 as well as severe frost and heat events. To*|

| S.No. | Reviewer's Comment | Authors' Response |
|---|---|---|
| | | *simulate crop yields (in decitons per hectare - dt/ha), we multiply the predicted yield anomaly by the district-level mean yield. This approach allows us to isolate crop yields attributable to hydro-meteorological extremes defined in Table 1, including droughts. These simulated yields are then used for damage assessment in drought-affected regions categorised using the SMI (as described in next section), aligning with the objective of quantifying the economic damages during droughts driven by the biophysical impacts of droughts and their interaction with other extremes. Descriptive statistics for the simulated yields, including their annual mean, minimum, and maximum values, are provided in Appendix A."* We have also updated the citation of the statistical yield model paper (Heilemann et al., 2024) from the pre-print to the published version that reflects the final, peer-reviewed publication. |
| 5. | The study simultaneously addresses two separate issues in its spatial disaggregation exercise. From my reading, the study disaggregates crops, as well as the country and it is not clear which of these is responsible for the differential when compared to national figures. This is especially true as the only differences come when crops are broken out and investigated individually. To summarize, would the national estimate lead to the same discrepancy without spatial disaggregation if the damage of each crop is investigated separately? (Basically, is the difference a result of disaggregating crops or spatial disaggregation) | Agreed and clarified: We appreciate the thoughtful comment and acknowledge that the original text may not have clearly conveyed the distinction between crop-specific and spatial disaggregation in damage estimates. To address this, we have clarified the mechanism leading to the discrepancy between nation level and the disaggregated assessment in the revised manuscript with the text below: Pg 10, lines 327-332: *"In our analysis, crop-specific damages are calculated both at the national level, using aggregated national data, and at the regional-level, using reported yields from each district. Regional-level damages are then summed to obtain national totals for comparison with aggregated national-level results. This approach allows us to compare the extent of differences in damage estimates between national-level and regional-level data sources while retaining a crop-specific focus in both cases, providing insights into the potential biases that may arise from relying solely on national-level data."* |

**Some typos…**

| S.No. | Reviewer's Comment | Authors' Response |
|---|---|---|
| 5. | Page 2 line 64: underestimates should be underestimate | Corrected and proofread: Thank you for thoroughly reviewing the manuscript and noting the typos. We have corrected the identified errors and carefully proofread the text to ensure no other such errors remain. |
| 6. | Page 2 line 77 "…are derived from a the…" delete "a" | |
| 7. | Page 3 line 97 "casual" should be "causal" | |

**Additional references, as suggested by the reviewers/editor, or included to address their feedback:**

Ahmad, M. M., Yaseen, M., & Saqib, S. E. (2022). Climate change impacts of drought on the livelihood of dryland smallholders: Implications of adaptation challenges. International Journal of Disaster Risk Reduction, 80, 103210. https://doi.org/10.1016/j.ijdrr.2022.103210

Badolo, F., & Somlanare, R. K. (2012). Rainfall shocks, food prices vulnerability and food security: Evidence for Sub-Saharan African Countries. Proceedings of the African Economic Conference, Kigali, Rwanda, 1.

Berhanu, M., & Wolde, A. (2019). Review on Climate Change Impacts and its Adaptation strategies on Food Security in Sub-Saharan Africa. Agricultural Social Economic Journal, 19, 145–154. https://doi.org/10.21776/ub.agrise.2019.019.3.3

Ding, J., Huang, Z., Zhu, M., Li, C., Zhu, X., & Guo, W. (2018). Does cyclic water stress damage wheat yield more than a single stress? PLoS ONE, 13(4), e0195535. https://doi.org/10.1371/journal.pone.0195535

Garbero, A., & Muttarak, R. (2013). Impacts of the 2010 Droughts and Floods on Community Welfare in Rural Thailand: Differential Effects of Village Educational Attainment. Ecology and Society, 18(4). https://doi.org/10.5751/ES-05871-180427

García-León, D., Standardi, G., & Staccione, A. (2021). An integrated approach for the estimation of agricultural drought costs. *Land Use Policy*, *100*, 104923. https://doi.org/10.1016/j.landusepol.2020.104923

Gray, M., Hunter, B., & Edwards, B. (2009). A Sunburnt Country: The Economic and Financial Impact of Drought on Rural and Regional Families in Australia in an Era of Climate Change. Australian Journal of Labour Economics (AJLE), 12, 108–131.

Heilemann, J., Klassert, C., Samaniego, L., Thober, S., Marx, A., Boeing, F., Klauer, B., & Gawel, E. (2024). Projecting impacts of extreme weather events on crop yields using LASSO regression. *Weather and Climate Extremes*, *46*, 100738. https://doi.org/10.1016/j.wace.2024.100738

Howitt, R., MacEwan, D., Medellín-Azuara, J., Lund, J., & Sumner, D. (2015). *Economic Analysis of the 2015 Drought For California Agriculture*. Center for Watershed Sciences, University of California – Davis.

*Hydroclimatic Forecasting System*. (2024). https://www.ufz.de/index.php?en=47304

Liu, S., Xiao, L., Sun, J., Yang, P., Yang, X., & Wu, W. (2022). Probability of maize yield failure increases with drought occurrence but partially depends on local conditions in China. *European Journal of*

*Agronomy*, *139*, 126552. https://doi.org/10.1016/j.eja.2022.126552

Lobell, D. B., Schlenker, W., & Costa-Roberts, J. (2011). Climate Trends and Global Crop Production Since 1980. Science, 333(6042), 616–620. https://doi.org/10.1126/science.1204531

Prasanna, R. P. I. R. (2018). Economic costs of drought and farmers' adaptation strategies: Evidence from Sri Lanka. Sri Lanka Journal of Economic Research, 5(2). https://doi.org/10.4038/sljer.v5i2.49

Ray, C. A. (2021). The Impact of Climate Change on Africa's Economies.
Suarez-Gutierrez, L., Müller, W. A., & Marotzke, J. (2023). Extreme heat and drought typical of an end-of-century climate could occur over Europe soon and repeatedly. Communications Earth & Environment, 4(1), 1–11. https://doi.org/10.1038/s43247-023-01075-y

van Duinen, R., Filatova, T., Geurts, P., & van der Veen, A. (2015). Coping with drought risk: Empirical analysis of farmers' drought adaptation in the south-west Netherlands. Regional Environmental Change, 15(6), 1081–1093. https://doi.org/10.1007/s10113-014-0692-y

Wens, M. L. K., Mwangi, M. N., van Loon, A. F., & Aerts, J. C. J. H. (2021). Complexities of drought adaptive behaviour: Linking theory to data on smallholder farmer adaptation decisions. International Journal of Disaster Risk Reduction, 63, 102435. https://doi.org/10.1016/j.ijdrr.2021.102435

Zipper, S. C., Qiu, J., & Kucharik, C. J. (2016). Drought effects on US maize and soybean production: Spatiotemporal patterns and historical changes. *Environmental Research Letters*, *11*(9), 094021. https://doi.org/10.1088/1748-9326/11/9/094021